# Afterglow ice formed by phosphorescent luminophore-protein conjugates and complexes in aqueous solution at freezing temperature

Xun Li, Jiuyang Li, Guangming Wang, Yuming Su, Minjian Wu & Kaka Zhang ✉

Unlike the widely reported fluorophore-biomacromolecule systems in nucleic acid testing and immunoassay, organic afterglow luminophore-biomacromolecule conjugates and complexes remain rarely explored. Here we report the observation of organic afterglow from aqueous solutions of luminophore-protein conjugates and complexes at freezing temperature, named as afterglow ice for abbreviation. Reliable N-hydroxysuccinimide ester protocol, as well as β-diketone chemistry, are applied for protein labeling to form luminophore-protein conjugates, which exhibit intriguing afterglow at freezing temperature. Control experiments reveal that hydrophobic interaction and covalent linkage between luminophore and protein can protect organic triplet excited states from quenching. In the case of luminophore-protein complex, we observe the switching-on of organic afterglow after specific recognition of streptavidin by biotinylated luminophore, which is the example of specific recognition and sensing of biomacromolecules by organic afterglow emitters. Although it is not yet a mature technology for biomedical applications, this study represents the initial step of organic afterglow materials towards bio-labeling and bioassay fields, as well as expanding the application scenarios of bioactive products.

Organic phosphorescent and afterglow materials, characterized by their long-lived luminescence, easily modifiable molecular structures, low toxicity, sustainability, biocompatibility, and cost-effectiveness, demonstrate significant promise for applications in biomedical imaging, information storage and anti-counterfeiting technologies, smart responsive materials and sensing detection, as well as optoelectronic devices[1–15]. In conventional organic systems, due to the spin-forbidden nature of singlet-to-triplet transition and weak spin-orbit coupling, the formation of triplet excited states is challenging and they are prone to deactivation at room temperature. Over the past decade, researchers have continually developed and refined design strategies for high-performance organic room-temperature phosphorescent (RTP) materials, focusing on aspects such as enhancing intersystem crossing, suppressing non-radiative deactivation, and modulating inter-molecular interactions[16–35]. It has been reported that organic molecular systems containing heavy atoms or carbonyl groups exhibit significantly enhanced intersystem crossing efficiency and improved phosphorescent quantum yields due to strong spin-orbit coupling effects, thereby showcasing exceptional room-temperature phosphorescent performance[17–23]. Additionally, the introduction of rigid microenvironments to constrain molecular vibrations has emerged as an effective strategy for inhibiting non-radiative deactivation

State Key Laboratory of Organometallic Chemistry and Shanghai Hongkong Joint Laboratory in Chemical Synthesis, Key Laboratory of Synthetic and Self-Assembly Chemistry for Organic Functional Molecules, Shanghai Institute of Organic Chemistry, University of Chinese Academy of Sciences, Chinese Academy of Sciences, 345 Lingling Road, Shanghai 200032, People's Republic of China. ✉e-mail: zhangkaka@sioc.ac.cn

processes[24–28]. Techniques such as constructing rigid molecular frameworks, crystal engineering, creating glassy environments, and employing two-component doping methodologies are commonly utilized[29–35]. Furthermore, by modulating intermolecular interactions, such as π-π stacking, hydrogen bonding, van der Waals forces, and supramolecular complexation, researchers can protect the triplet excited state, consequently enhancing the phosphorescent emission efficiency and stability of the materials[36–41]. Through manipulation of excited states, a variety of organic RTP systems with excellent phosphorescent properties and diverse functionalities have been successfully developed[1–15,42,43].

In addition to advanced anti-counterfeiting and information encryption applications[1–15], researchers have recently leveraged the environment-sensitive characteristics of RTP materials to develop smart sensors. These materials enable the detection of external environmental changes, such as chemical stimuli, illumination, temperature, pH, moisture, and mechanical force, by monitoring variations in luminescence intensity, lifetime, or spectral properties without interference from background signals, demonstrating functionalities in chemical sensing, gas detection, water quality analysis, and environmental pollutant monitoring[44–60]. Literature reports have shown that supramolecular inclusion-based RTP systems exhibit selectivity towards luminescent guests; for instance, the degree of RTP enhancement in solid-state crown ether RTP systems varies upon the association with different metal ions[56]. Similarly, a reported study showed that organic afterglow materials dispersed in aqueous environments can diminish or quench in the presence of ferric and ferrous ions[59]. Despite these intriguing advancements, there have been few report of responsive RTP systems capable of potentially analyzing and detecting biomacromolecules, nor studies focused on their potential specific recognition of biomacromolecules. In contrast, in the application fields of nucleic acid testing and immunoassays, organic fluorescent molecules can effectively cooperate with precisely structured biomacromolecules, either through chemical conjugation or non-covalent binding, leveraging the specificity of biomacromolecule recognition alongside the optical reporting capabilities of organic fluorescent molecules to achieve high-value biomedical applications[61,62]. Given this context, we believe that the construction of conjugates and complexes between biomacromolecules and afterglow molecules will be critically important for advancing the biomedical applications of these luminescent systems, particularly given that afterglow materials exhibit a significantly longer luminescence lifetime —an attribute that is essential for high-contrast bioimaging and

sensitive bioanalysis. Previous studies showed the covalently linking of bromo-substituted naphthalimide derivative to bovine serum albumin and the measurement of the RTP properties of the resulting protein-dye conjugate[63]. Our recent work has doped difluoroboron β-diketonate ($BF_2$bdk) molecules into matrices such as silk cocoons, shells, and feathers to develop biomass-based afterglow materials[64]. These investigations have primarily examined afterglow properties in solid-state forms[63,64], with no reported studies on luminophore-protein conjugates or complexes in solution or dispersion states.

Based on our research experience and literature reports, we find that the realization of afterglow emission from organic molecules in solution or dispersion states is inherently challenging due to the typically low afterglow emission rate constant for organic phosphorescent molecules (for afterglow lifetimes exceeding 100 ms, the afterglow rate constant should be <10 s$^{-1}$), in contrast to the significantly higher non-radiative deactivation and oxygen quenching rates in these environments, often reaching $10^2$ to $10^3$ s$^{-1}$ or even greater. In biological system, there exists an enzyme named horse liver alcohol dehydrogenase whose tryptophan residue (Trp 314 in the sequence) can absorb ultraviolet light to form a triplet excited state ($T_1$) through intersystem crossing, and the enzyme's rigid microenvironment protects this $T_1$ state, enabling afterglow emissions at room temperature with a phosphorescence lifetime around 0.1 s[65]. Inspired by this, we propose that if organic luminophore were covalently linked to or complexed with proteins, certain regions of the protein might protect the luminophore's $T_1$ states. Furthermore, by appropriately lowering the temperature to suppress non-radiative deactivation and oxygen quenching of $T_1$ excited state, such a luminophore-protein conjugate or complex system may enable afterglow emissions at ambient or conventional refrigerator temperatures.

Here we select proteins with hydrophobic regions and hydrophobic luminescent molecules to construct organic luminophore-protein conjugates and complexes in aqueous phases through chemical conjugation and non-covalent binding, respectively (Fig. 1). Driven by hydrophobic interactions, the luminescent molecules would associate with the hydrophobic microregions of the proteins. In the investigation of the organic luminophore-protein conjugates, we initially synthesized luminescent molecules containing N-hydroxysuccinimide (NHS) ester groups and employed bovine serum albumin (BSA) as a model protein, utilizing the traditional NHS ester labeling method to study this system. The resulting aqueous solution of the luminophore-BSA conjugates exhibited significant organic afterglow at freezing point (that is the afterglow ice), while free

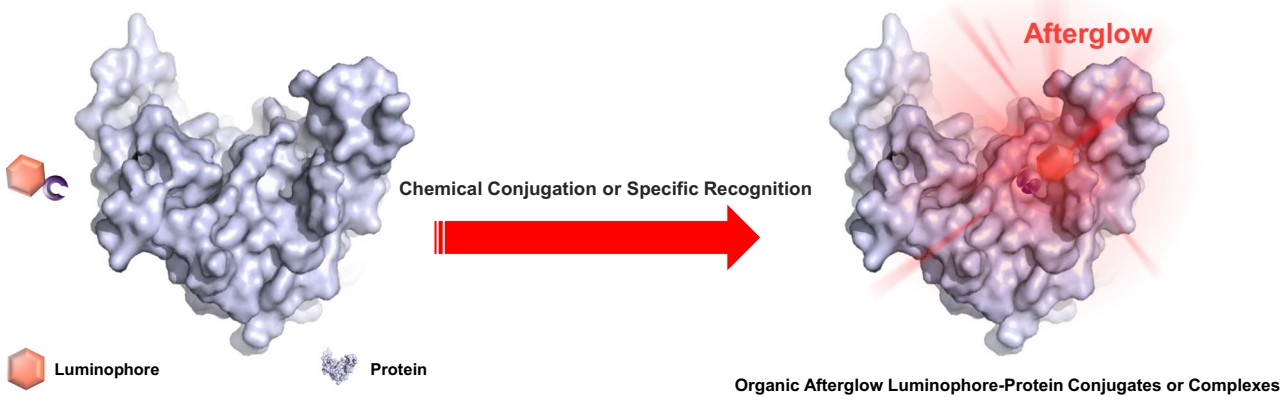

**Luminophore**   **Protein**

**NHS, β-diketone in covalent system or Biotin in specific recognition system**

**Afterglow**

**Chemical Conjugation or Specific Recognition**

**Organic Afterglow Luminophore-Protein Conjugates or Complexes**

**Fig. 1 | Formation of luminophore-protein conjugates and complexes via covalent linkage and non-covalent specific recognition, respectively, as well as the observation of organic afterglow from the conjugates and complexes at** freezing temperature. The protein cartoons in this study are prepared using publicly available protein CIF files (https://www.rcsb.org) and PyMOL software (http://www.pymol.org/pymol).

luminescent molecules and several other control samples showed no notable afterglow under the same conditions. In addition, we introduced a benzophenone functional group to enhance the intersystem crossing of the luminescent molecules, leading to stronger afterglow from the luminophore-BSA conjugates at freezing point. Moreover, we developed a β-diketone-based protein labeling method that achieved bright afterglow across the full color spectrum for organic luminophore-BSA conjugates. We subsequently modified the surface of polymer microspheres with luminophore-protein conjugates to obtain afterglow protein microspheres. These microspheres inherit the properties of afterglow luminophore and proteins, thereby offering potential applications in bio-analysis and bio-detection. In our study of organic luminophore-protein complexes, we utilized the classic streptavidin (SA) and biotin system (with a binding constant on the order of $10^{15}$ M$^{-1}$) for proof-of-concept demonstration. The synthesized biotinylated luminophore specifically binds to SA, resulting in a complex that exhibits visible phosphorescent afterglow at freezing temperature. This indicates that the specifically modified phosphors have potential for the specific recognition of proteins. Besides, we perform photophysical studies of luminophore-protein complexes in bile acid systems and demonstrate that the afterglow ice phenomenon provides a visual pathway to identify target proteins for specific biologically active compounds.

## Results

### Afterglow Luminophore-Protein Conjugates Obtained by NHS Ester Chemistry

We selected BSA as a model protein due to its abundant hydrophobic microregions, which allow for reversible binding with various substances, particularly those with low water solubility[66]. The naphthalene group, which is hydrophobic, is a well-known phosphorescent moiety; here we chose methoxynaphthalene as phosphorescent unit. By chemical modification, we synthesized **1NAP-NHS** to label BSA, subsequently obtaining an aqueous solution of **1NAP-BSA** conjugates after extensive dialysis against deionized water (Fig. 2). The average molecular weight of **1NAP-BSA** conjugates by matrix-assisted laser desorption/ionization time-of-flight (MALDI-TOF) mass spectrometry (66960) exceeds that of BSA (66531) by 429 Da (Supplementary Fig. 1), corresponding to the addition of approximately two **1NAP** group. These results agree with the feeding molar ratio of **1NAP-NHS**/BSA (2/1) and the estimated labeling efficiency (95.6%) (Supplementary Fig. 2). It is known that MALDI-TOF works at relatively high energy, where weak bonds and non-covalent binding would be disrupted, so the molecular increase by 429 Da can be unambiguously attributed to the covalent linkage of **1NAP** group to BSA through NHS ester chemistry. The solution of **1NAP-BSA** conjugates after dialysis was stored in the freezer (−20 °C) of a refrigerator (Fig. 2a), and it was observed that the frozen **1NAP-BSA** conjugates emitted yellow afterglow when excited by a handheld UV lamp at 365 nm (Fig. 2c). Notably, the afterglow emission remained significant when the frozen **1NAP-BSA** conjugates turned into an ice-water mixture (0 °C).

UV-Vis absorption analysis of the **1NAP-BSA** conjugates after dialysis revealed an absorption band in the range of 300–350 nm (Fig. 2b), which can be exclusively assigned to **1NAP** group, confirming the successful labeling of BSA with **1NAP** group. Additionally, photoluminescence spectroscopy conducted on the frozen **1NAP-BSA** conjugates showed a delayed emission peak at 550 nm under 365 nm excitation (Fig. 2c), which closely matched the delayed emission spectrum of **1NAP** when doped in the small-molecule organic matrix such as benzoic acid (Supplementary Fig. 3). The average phosphorescence lifetime ($\tau_{AVE}$) of the frozen **1NAP-BSA** conjugates was measured to be 290 ms (Fig. 2d). To further elucidate the origin of phosphorescence, several control experiments were performed. Firstly, a dialyzed solution of pristine BSA was prepared following identical experimental protocols but without the addition of **1NAP-**

**NHS**, and no afterglow was observed at −20 °C upon 365 nm excitation (Supplementary Fig. 4). Furthermore, both high-concentration aqueous BSA solution and BSA powder exhibited no afterglow at −20 °C under 365 nm excitation, although the BSA powder displayed faint and noisy delayed emission spectra (Supplementary Fig. 5). Moreover, for **1NAP-PEG** aqueous solution at 15 mg/mL (**1NAP** is not soluble in water, so we use polyethylene glycol (PEG) to modify **1NAP** into **1NAP-PEG**, Supplementary Information), we found that the frozen **1NAP-PEG** solution (**1NAP** group, 1.86 mg/mL) show fluorescence emission but no afterglow under 365 nm excitation (Supplementary Fig. 6a), contrasting sharply with the evident afterglow emitted by the frozen **1NAP-BSA** conjugates at a low concentration (BSA, around 1 mg/mL; **1NAP** group, about 0.024 mg/mL) (Supplementary Fig. 6b), thereby underscoring the importance of the hydrophobic microregions in proteins. Finally, upon non-covalently loading **1NAP** onto BSA, the frozen solution at −20 °C yielded very weak afterglow under 365 nm excitation (Supplementary Fig. 7). This experiment not only elucidates that the observed bright afterglow does not arise from a simple blending of luminescent molecules and proteins but also underscores the necessity of covalent linkage between **1NAP** and BSA. Through these control experiments, we propose that, driven by hydrophobic interactions, **1NAP-NHS** can bind to some hydrophobic microregion of BSA and then form conjugates with BSA via amide bonds. The synergistic effects of hydrophobic interactions and covalent bonding protect **1NAP**'s triplet excited state, thereby enabling afterglow emission from the conjugates at conventional refrigerator temperatures.

Nuclear magnetic resonance (NMR) technique is used to study the mobility of luminophore in different microenvironments. When **1NAP** groups are covalently linked to PEG or ε-polylysine (EPL) chains, the resultant **1NAP-PEG** and **1NAP-EPL** samples in D$_2$O show sharp and well-resolved $^1$H NMR signals in the aromatic region (Supplementary Figs. 8, 9). The **1NAP** groups have free motion in D$_2$O to give sharp peaks, because the hydrophilic polymers (PEG and **EPL**) endow **1NAP** groups with excellent solubility in D$_2$O. In contrast, $^1$H NMR spectrum of **1NAP-BSA** conjugates in D$_2$O shows the disappearance of NMR signals of **1NAP** groups in the aromatic region (Supplementary Fig. 10), which suggest the motion of **1NAP** groups is seriously restricted within BSA[67]. The hydrophobic **1NAP** groups within some hydrophobic microregion of BSA have largely restricted motion, which can significant reduce nonradiative decay of **1NAP**'s triplet excited states, leading to the emergence of afterglow at freezing temperature.

Subsequently, we selected various phosphorescent moieties including acridone, pyrene, naphthalene, benzophenone, biphenyl, and fluorene for labeling BSA via NHS ester chemistry, resulting in the formation of diverse luminophore-BSA conjugates (the chemical structures of corresponding **Ar-NHS** compounds can be found in Fig. 2e). The labeling efficiency estimated by UV-vis technique is >95% (Supplementary Fig. 2). Among these, the **Acridone-BSA** conjugates displayed bright blue-green afterglow and revealed a dual emission in its delayed spectrum (Fig. 2f, g). Based on our previous studies on acridone derivatives[68,69], the dual emission can be assigned as thermally activated delayed fluorescence and phosphorescence. The **Pyrene-BSA** conjugates demonstrated a vivid red afterglow with phosphorescence maximum at 620 nm (Fig. 2h) and an average afterglow lifetime of 188 ms (Supplementary Fig. 11). The frozen **2NAP-BSA** conjugates exhibited strong yellow-green afterglow emission and an average afterglow lifetime of 265 ms (Supplementary Fig. 12). The **BP-BSA, MeOBip-BSA**, and **Fluorene-BSA** conjugates exhibited relatively weak afterglow emissions (Supplementary Figs. 13, S14).

### Bright afterglow conjugates via ArBP-NHS labeling of proteins

To enhance the afterglow performance of luminophore-BSA conjugates, we introduced a benzophenone functional (BP) group into the **Ar-NHS** system to form **ArBP-NHS** compounds (Fig. 3). Benzophenone is recognized as a significant phosphorescent emitter, characterized

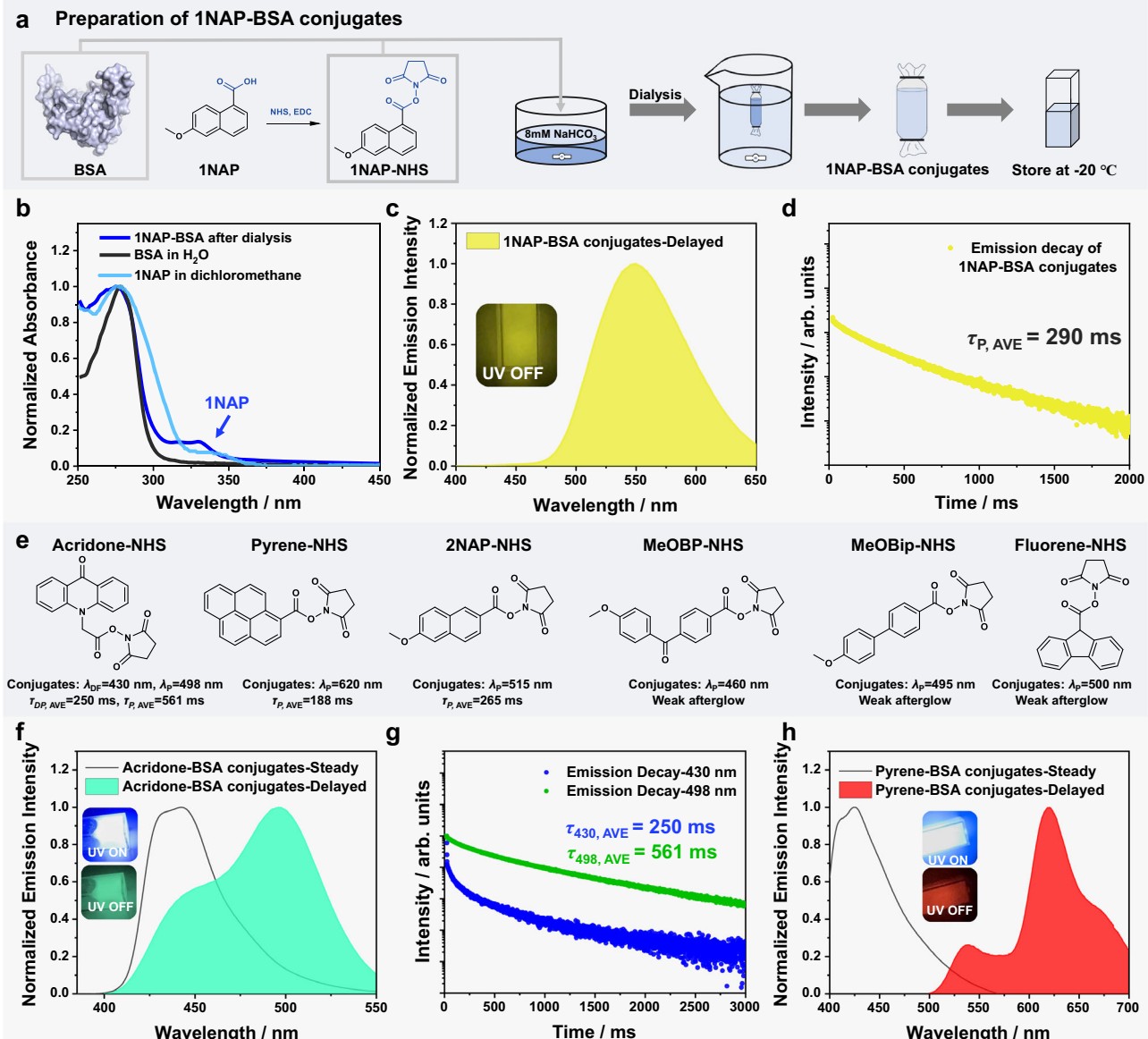

**Fig. 2 | Afterglow luminophore-protein conjugates obtained by NHS ester chemistry. a** Preparation of **1NAP-BSA** conjugates. **b** UV-vis absorption spectra of the aqueous solution of **1NAP-BSA** conjugates after dialysis, BSA aqueous solution and **1NAP** in dichloromethane solution. **c** Delayed emission spectrum of **1NAP-BSA** conjugates at conventional refrigerator temperature (in the freezer, −20 °C) under 365 nm excitation and **d** the corresponding emission decay curve of **1NAP-BSA** conjugates. $\tau_{P, AVE}$ denotes average phosphorescence lifetime. **e** Chemical structures of **Ar-NHS** compounds. $\lambda_{DF}$ and $\lambda_P$ represent emission maxima of delayed fluorescence and phosphorescence, respectively. **f** Steady-state and delayed emission spectra of the frozen **Acridone-BSA** and **g** the corresponding emission decay curve. **h** Steady-state and delayed emission spectra of the frozen **Pyrene-BSA** conjugates under 365 nm excitation. Source data are provided as a Source Data file.

by an intersystem crossing efficiency that approaches unity. However, it exhibits a notably short phosphorescence lifetime, typically ranging from 0.1 to 1 ms, primarily due to the pronounced n-π* transition inherent to its triplet state. Recent research indicates that advanced charge transfer technology can maintain strong intersystem crossing in benzophenone-derived organic systems and simultaneously reduce phosphorescence rate constant ($k_P$) to small values[23,70]. When non-radiative decay pathways are restricted using a rigid environment, it becomes possible to achieve highly efficient and persistent room-temperature phosphorescence. The benzophenone moiety is electron-deficient. By connecting it with electron-donating aromatic groups, an intramolecular charge transfer (ICT) system is established. The donor group contributes actively to the triplet excited state formation of the resulting system. If the donating group possesses a low-energy $T_1$ state of localized excitation (LE) character, the $T_1$ state of the resulting

molecules would consist of significant portion from the donating group's $T_1$ state. As a result, the resulting molecules' $T_1$ states would inherit the donating group's LE character and thus have small $k_P$. Naphthoic acid features a relatively low $T_1$ energy level; upon conjugation with the BP group, the resultant **NAPBP-COOH** molecule capitalizes on the localized excitation characteristics of naphthoic acid's triplet state, thereby facilitating the formation of a long-lived $T_1$ state within **NAPBP-COOH**[23,70]. This molecular design effectively preserves the robust intersystem crossing capability of the benzophenone framework while attenuating the contribution of the n-π* transition to the $T_1$ state. Consequently, this configuration endows the luminophore with both high intersystem crossing efficiency and a small phosphorescence rate constant. The efficacy of this molecular design is corroborated by experimental data from small-molecule doping systems, which demonstrate bright phosphorescence with an emission

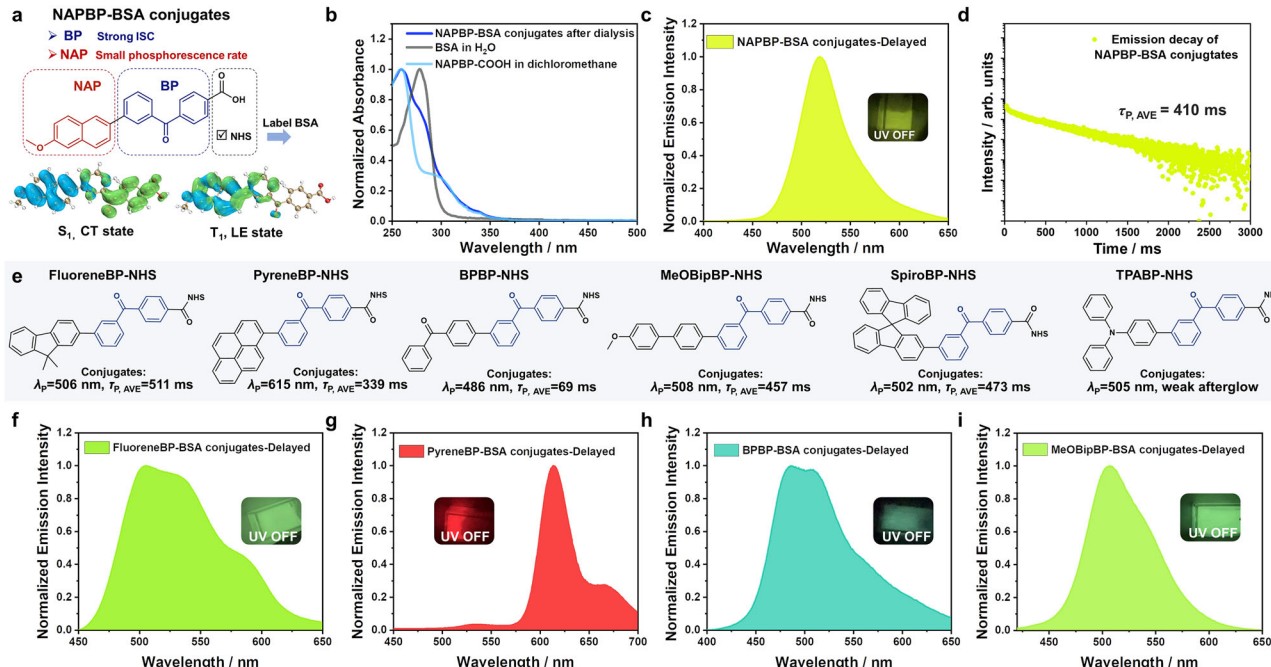

**Fig. 3 | Bright afterglow conjugates via ArBP-NHS labeling of proteins.**
**a** Molecular design of **NAPBP** system. **b** UV-vis absorption spectra of the **NAPBP-BSA** conjugates, BSA aqueous solution, and **NAPBP-COOH** in dichloromethane solution. **c** Delayed emission spectrum of the frozen **NAPBP-BSA** conjugates under 365 nm excitation and **d** the corresponding emission decay curve. **e** Chemical structures of **ArBP-NHS** compounds. **f–i** Delayed emission spectra of the frozen **FluoreneBP-BSA** (f), **PyreneBP-BSA** (g), **BPBP-BSA** (h) and **MeOBipBP-BSA** (i) conjugates under 365 nm excitation. Source data are provided as a Source Data file.

maximum/shoulder at 527/555 nm and a phosphorescence lifetime extending up to 524 ms (Supplementary Fig. 15). Consequently, we utilized **NAPBP-NHS** to label BSA, obtaining **NAPBP-BSA** conjugates. The UV-vis absorption spectrum of the dialyzed solution revealed the presence of an absorption band corresponding to **NAPBP** (Fig. 3b). The aqueous solution of the conjugates at room temperature shows negligible delayed emission signals (Supplementary Fig. 16), which can be explained by the active nonradiative decay and oxygen quenching of luminophore's triplet excited states at fluidic aqueous solution. After storing the dialyzed solution in a refrigerator (−20 °C), the frozen **NAPBP-BSA** conjugates under 365 nm excitation displayed phosphorescence maximum/shoulder at 526/559 nm in the delayed emission spectrum (Fig. 3c). The average afterglow lifetime was measured to be 410 ms (Fig. 3d). In comparison to the **2NAP-BSA** conjugates (without BP attachment), the phosphorescence intensity and lifetimes are significantly enhanced in the system of **NAPBP-BSA** conjugates; under excitation by a 365 nm handheld UV lamp, a brighter and longer afterglow emission was visually observed in **NAPBP-BSA** system (Fig. 3c). We collect temperature-dependent delayed emission spectra and emission decay profiles for the **NAPBP-BSA** conjugates. The delayed emission intensity is very weak at 278 K because the active nonradiative decay and oxygen quenching of triplet excited states in the fluidic aqueous solution. Upon freezing, the delayed emission intensity has significant increase (Supplementary Fig. 17a), indicating the essential role of ice for the protection of **NAPBP**'s triplet excited states. It has also been found that, by further decreasing the temperature, the delayed emission intensity and phosphorescence lifetimes increase (Supplementary Fig. 17b), which can be attributed to better protection of **NAPBP**'s triplet excited states within the ice at lower temperature. Time-dependent density functional theory (TD-DFT) calculations indicated that the $S_1$ excited state of **NAPBP-COOH** displayed typical characteristics of intramolecular charge transfer, there are rich ISC channels with substantial spin-orbit coupling matrix element (SOCME) values, and the $T_1$ excited state exhibited features indicative of significant localized excitation (Supplementary Fig. 18).

These findings suggest that the **NAPBP** system possesses strong ISC along with a slow phosphorescence decay rate, which is consistent with our experimental observation.

We understand that some compounds may precipitate at low temperature that may be responsible for the emergence of afterglow in the present system. To study this, aqueous solution of **NAPBP-BSA** conjugates are immediately frozen by liquid nitrogen and then placed in −20 °C refrigerator for 1 day. The sample shows homogenous afterglow upon removing excitation source (Supplementary Fig. 19). Besides, the **NAPBP-BSA** conjugates have been diluted to 1 mg/mL. The frozen solution of this diluted sample still shows significant afterglow (Supplementary Fig. 20A–C). Since the extent of precipitation or phase separation would be relatively low during rapid freezing and for samples at low concentration, these results suggest the emergence of afterglow in the present study is unlikely caused by precipitation or phase separation. To further study whether protein precipitation/phase separation during freezing contributes to the emergence of afterglow, we covalently link PEG chains to **NAPBP-BSA** conjugates; the steric repulsion between PEG chains would minimize the possibility of protein aggregation especially in the case when the **NAPBP-BSA** conjugates are heavily decorated with PEG chains. After being frozen, the PEG-decorated **NAPBP-BSA** conjugates have been found to show similar afterglow properties compared to **NAPBP-BSA** conjugates (control sample, without PEG) (Supplementary Fig. 20D–Q). These studies show that protein aggregation may occur during freezing in the case of luminophore-protein conjugates but has insignificant contribution to afterglow emergence in the present system, and thus support our proposed mechanism of synergistic effects of conjugate formation and hydrophobic interactions at freezing temperature for the emergence of afterglow.

It should be noted that, in the **NAPBP-BSA** system, the $T_1$ states of localized excitation character with very small oscillator strength would have limited tendency to undergo photochemical reaction with protein. Besides, **NAPBP-BSA** conjugates at low concentrations (2.5 mg/mL to 7.5 mg/mL) have negligible or low cytotoxicity to NIH3T3 cells

(Supplementary Fig. 21). The cytotoxicity increases with sample concentration but is still acceptable at higher concentrations (10 mg/mL, Supplementary Fig. 21).

By substituting the **NAP** group with other functional groups, such as dimethylfluorene, pyrene, benzophenone, biphenyl, spirobifluorene, and triphenylamine, we synthesized six additional **ArBP-NHS** molecules (Fig. 3e). After BSA labeling, all six **ArBP-BSA** conjugates exhibited persistent afterglow emission at freezing temperature. Notably, under 365 nm excitation, four of the systems demonstrated afterglow lifetimes exceeding 300 ms (Supplementary Figs. 22, S23). The **FluoreneBP-BSA** conjugates emitted a bright bluish-green afterglow (Fig. 3f), with an average phosphorescence lifetime of 511 ms (Supplementary Fig. 22a), while the **PyreneBP-BSA** conjugates emitted a vivid red afterglow at 615 nm (Fig. 3g), with an average phosphorescence lifetime of 339 ms (Supplementary Fig. 22b). In the examples of **NAPBP-BSA, PyreneBP** and **MeOBipBP**, the introduction of the BP group significantly enhanced the afterglow of the luminophore-BSA conjugates, when compared to those in **2NAP-BSA, Pyrene-BSA** and **MeOBip-BSA** systems (Fig. 3c, g, i). These results further validate the effectiveness of the **ArBP-NHS** molecular design. Photophysical data for other **ArBP-BSA** conjugates are detailed in Fig. 3h, Supplementary Figs. S23, S24.

## Interesting β-Diketone-based covalent labeling of proteins: a serendipitous discovery

Building upon the above NHS ester chemistry, we aimed to label BSA with our phosphorescent difluoroboron β-diketonate (**BF₂bdk**) molecules such as **2COOHNAPBF₂-NHS**, hypothesizing that this would yield robust persistent luminescence (Fig. 4a). The results demonstrated that the obtained luminophore-protein conjugates at freezing point emitted an expected bright yellow-green long afterglow. Surprisingly, control experiments revealed that treatment of **2COOHNAPBF₂** (without NHS ester) with BSA under identical conditions also resulted in intense yellow-green afterglow emission (Fig. 4b). Initially, we speculated that this phenomenon arises from a strong non-covalent interaction between **2COOHNAPBF₂** and BSA; the combination of hydrogen bonding and hydrophobic interaction between **2COOHNAPBF₂** and BSA might be responsible for this strong non-covalent interaction. In control experiment, we removed the carboxyl group of **2COOHNAPBF₂** to form **0COOHNAPBF₂**, and subjected it to the same experimental procedures (stirring with BSA, dialysis, and being stored at −20 °C). We still observed bright afterglow emission, thus ruling out the contribution of carboxyl group/hydrogen bonding for the emergence of afterglow. We then proposed an alternative mechanism whereby, in alkaline environments, the **BF₂** group of the phosphorescent molecule readily dissociates. The resulting β-diketone moiety can react with the amine groups of proteins to form β-enaminone compounds (Fig. 4c)[71,72], suggesting that the afterglow in this system is induced by the covalent linkage between β-diketone and BSA. The ice matrix also plays essential role for the emergence of afterglow as evidenced by the temperature-dependent delayed emission spectra and emission decay profiles, where the phosphorescence intensity and lifetimes increase upon freezing and lowering temperature (Supplementary Fig. 25). In control experiment, the aqueous solution of the conjugates at room temperature has negligible delayed emission signals because of the active nonradiative decay and oxygen quenching in fluidic aqueous medium (Supplementary Fig. 26). Besides temperature, we also investigate the contribution of hydrophobic effect on the afterglow property. Sodium dodecyl sulfonate (SDS) is known to disrupt hydrophobic interactions in protein systems. We add SDS into the aqueous solution of **2COOHNAP-diketone-BSA** conjugates. After being stored at −20 °C, the frozen sample with the presence of SDS shows much weaker phosphorescence than the frozen sample without SDS (Supplementary Fig. 27). This observation supports our proposed mechanism that hydrophobic interaction and

covalent linkage between luminophore and protein can protect organic triplet excited states from quenching.

To validate this proposed β-diketone-based covalent labeling chemistry, we divided **2COOHNAPBF₂** into five groups under alkaline conditions: one untreated group and the remaining groups treated with various amines, including glycine (Gly), L-lysine (Lys), L-arginine (L-Arg), and n-butylamine. High-performance liquid chromatography with mass spectrometry (HPLC-MS) was employed to monitor the formation of β-diketone and β-enaminone products. The HPLC-MS results indicated that free **2COOHNAPBF₂** generates a β-diketone with a m/z of 287 in mildly alkaline conditions (Supplementary Fig. 28). Additionally, β-diketone peaks and corresponding β-enaminone peaks were observed for all four amine-treated systems (Supplementary Figs. 29, S30), confirming that β-diketones can react with amine groups to produce β-enaminones. Moreover, we aimed to further establish the feasibility of β-diketone-based protein labeling through SDS-PAGE electrophoresis. Following reactions of five **BF₂bdk** compounds with BSA in alkaline conditions and subsequent dialysis, we obtained five **BF₂bdk**-BSA aqueous solutions exhibiting fluorescence when excited at 365 nm. These were tested alongside a control group of pristine BSA in SDS-PAGE. All six samples exhibited similar migration patterns under daylight. Under 365 nm excitation, the **BF₂bdk**-BSA solutions showed fluorescence, while the pristine BSA didn't exhibit any fluorescence (Fig. 4d and Supplementary Fig. 31). This result corroborates the ability of β-diketones to achieve covalent labeling of proteins.

Consequently, we aim to develop a β-diketone-based protein labeling strategy that triggers organic afterglow in luminophore-protein conjugates. As illustrated in Fig. 4c, we synthesized the **2COOHNAP-diketone** molecule and labeled it with BSA. UV-Vis absorption spectral data from the dialyzed solution of **2COOHNAP-diketone-BSA** confirmed successful labeling of BSA by the β-diketone (Fig. 4e). Upon refrigeration, the dialyzed solution emitted a bright yellow-green afterglow when excited by a 365 nm handheld UV lamp, and its delayed emission spectrum revealed a distinct phosphorescence band of **2COOHNAP-diketone** (Fig. 4f), with an average afterglow lifetime of 201 ms (Fig. 4g), which is supported by experimental data from small-molecule doping systems (Supplementary Fig. 32). Furthermore, we conducted a control experiment without the presence of BSA, modifying **2COOHNAP-diketone** with hydrophilic polymer PEG and preparing an aqueous solution at 13 mg/mL (**2COOHNAP group**, about 2.02 mg/mL), which was subsequently frozen in the refrigerator; under irradiation with a 365 nm UV lamp, no afterglow emission was detected (Supplementary Fig. 33). In contrast, the **2COOHNAP-diketone-BSA** conjugates exhibited significant afterglow behavior even at 1 mg/mL (**2COOHNAP group**, about 0.012 mg/mL), underscoring the importance of BSA's hydrophobic microregion for the emergence of afterglow. Integrating the results that the frozen solution of single-component BSA showed no afterglow (Supplementary Fig. 4), we propose that **2COOHNAP-diketone** interacts with some hydrophobic microregion of BSA through hydrophobic interactions. This interaction facilitates the formation of luminophore-protein conjugates via reactions between the β-diketone and amino groups of the protein. As a result of the synergistic effect of hydrophobic interactions and covalent bonding, the triplet excited state of the luminophore is protected, enabling phosphorescent emission of the conjugates at freezing temperatures. The **2COOHNAP-diketone-BSA** conjugates show negligible or low cytotoxicity to NIH3T3 cells at low concentrations (2.5 mg/mL to 7.5 mg/mL) and acceptable cytotoxicity at a higher concentration (10 mg/mL, Supplementary Fig. 34).

Given that the **BF₂bdk** molecules can readily convert into β-diketone compounds in alkaline condition, more **BF₂bdk** molecules with diverse structures have been synthesized via our cascade reaction (Fig. 5) for covalent labeling of BSA. All 19 **BF₂bdk** molecules can successfully label BSA, resulting in conjugates that exhibited full-spectrum afterglow emission at freezing point. By UV-vis

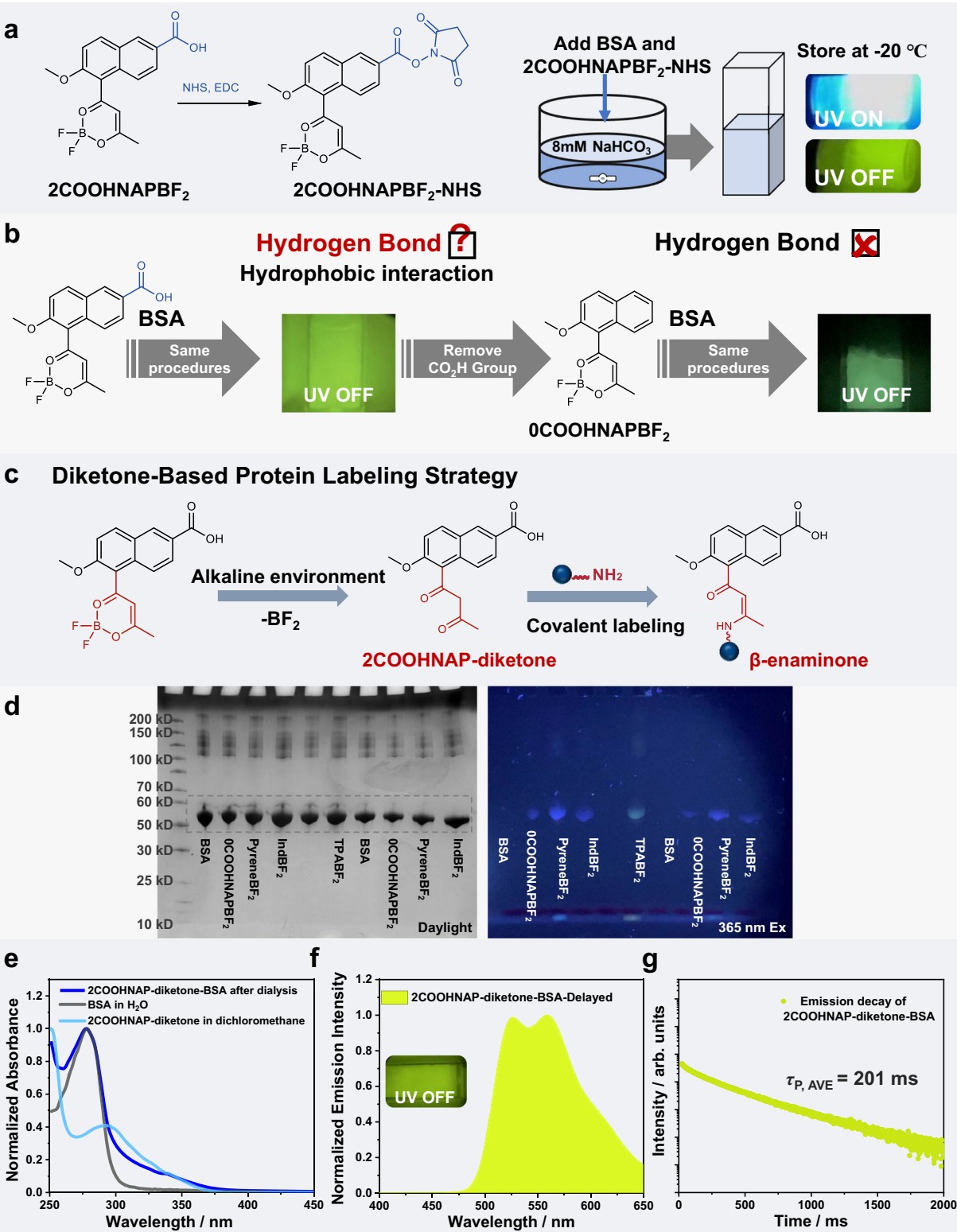

**Fig. 4 | β-Diketone-based covalent labeling of proteins. a** Synthesis of **2COOHNAPBF₂-NHS** for BSA labeling and the photographs of the obtained afterglow ice of conjugates solution. **b** Two sets of control experiments utilizing **2COOHNAPBF₂** and **0COOHNAPBF₂** as phosphorescent molecules. **c** The interesting protein labeling method based on the reaction between β-diketone and amine to form β-enaminone. **d** SDS-PAGE analysis of BSA and **BF₂bdk**-BSA conjugates. Uncropped and unprocessed scans of gels were added in Source Data file.

Four times each experiment was repeated independently with similar results. **e** UV-vis absorption spectra of the **2COOHNAP-diketone-BSA** conjugates after dialysis, BSA aqueous solution, and **2COOHNAP-diketone** in dichloromethane solution. **f** Delayed emission spectrum of the frozen **2COOHNAP-diketone-BSA** conjugates upon excitation at 365 nm and (**g**) the corresponding emission decay curve monitored at 520 nm. Source data are provided as a Source Data file.

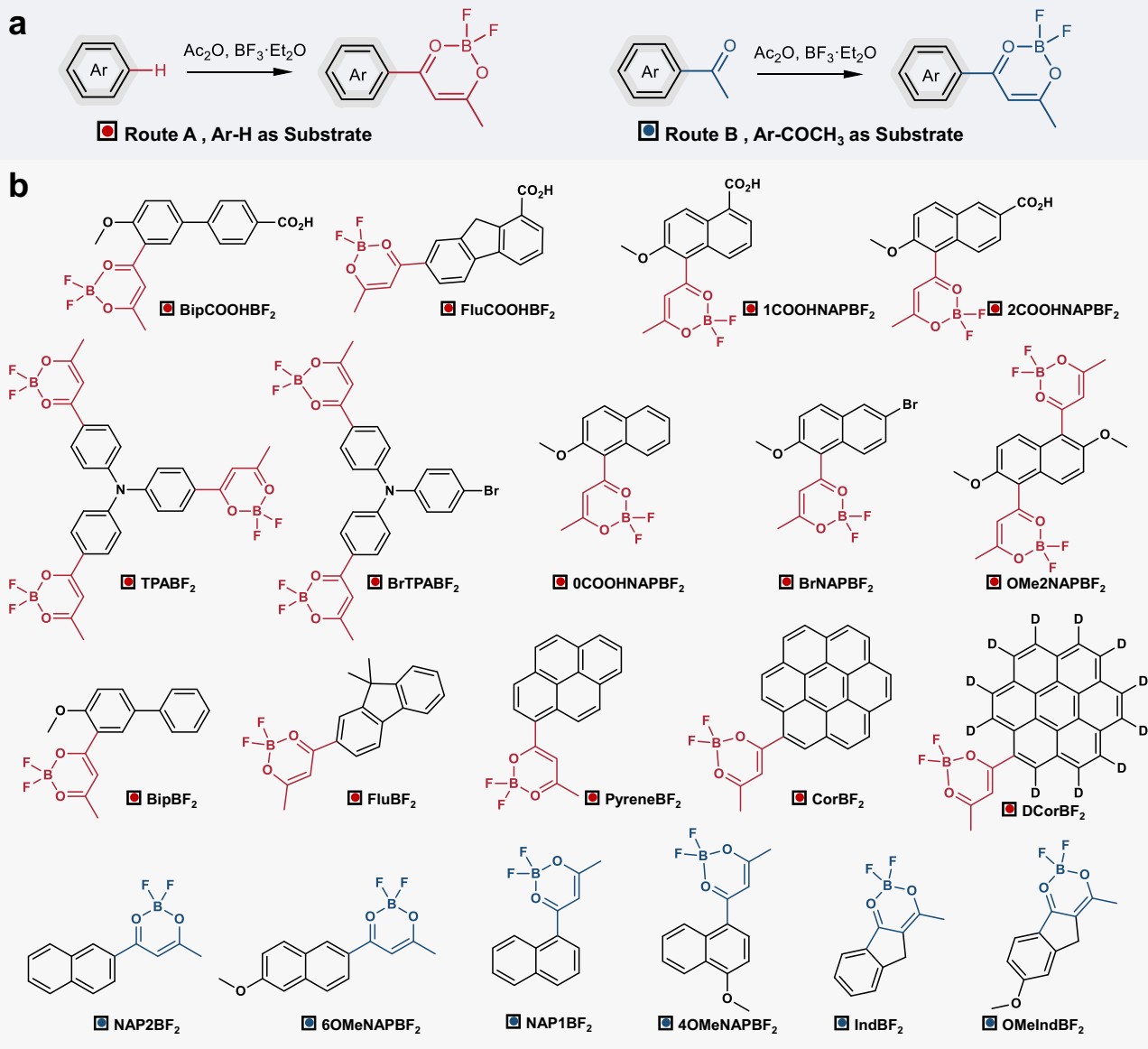

**Fig. 5 | BF₂bdk molecules for protein labeling. a** Two synthetic routes for **BF₂bdk** molecules. **b** Chemical structures of diverse **BF₂bdk** molecules, where red and blue symbol represents molecules synthesized from Route A and B, respectively.

measurement, the labeling efficiency can be estimated to be 23.1% to 89.5% (Supplementary Fig. 35). Notably, upon excitation at 365 nm, the **BipCOOHBF₂-BSA** conjugates (Fig. 6a), **OMeIndBF₂-BSA** conjugates (Fig. 6b), **IndBF₂-BSA** conjugates (Fig. 6c), **BrTPABF₂-BSA** conjugates (Fig. 6d), **1COOHNAPBF₂-BSA** conjugates (Fig. 6e), and **PyreneBF₂-BSA** conjugates (Fig. 6f) demonstrated bright blue, cyan, green, yellow-green, yellow, and red phosphorescence, respectively, with wavelengths ranging from 477 to 630 nm. The emission decay curve corresponding to the six **BF₂bdk-BSA** conjugates are presented in Supplementary Fig. 36. Moreover, the steady-state and delayed spectra of the **BrTPABF₂-BSA** and **TPABF₂-BSA** conjugates overlapped almost completely, resulting in afterglow emission based on the TADF mechanism (Fig. 6d and Supplementary Fig. S37). The **CorBF₂-BSA** conjugates exhibited a long phosphorescence lifetime of up to 600 ms (Supplementary Fig. 38). We apply deuteration technique to increase phosphorescence lifetimes of luminophore-protein conjugates. C-D stretching vibration is much weaker than C-H stretching vibration. Deuteration of luminophores can reduce the energy of effective non-classical vibrational mode involved in the nonradiative transition, decrease Franck-Condon factor, and consequently reduce

nonradiative decay, leading to the elongation of phosphorescence lifetime[24]. Deuterated CorBF₂ (**DCorBF₂**) has been prepared and used to label BSA; the synthetic details can be found in Supplementary Information. The resultant aqueous solution **DCorBF₂-BSA** conjugates at frozen state display phosphorescence lifetime of 2.29 s (Fig. 6g, h), which remains very rare in the related afterglow material systems (Supplementary Fig. 39). Supplementary Fig. 39 summarizes the reported afterglow materials in aqueous systems including supramolecular assemblies, suspension, emulsion and hydrogels, from which one can find that it is challenging to prepare materials with long phosphorescence lifetimes because of the active nonradiative decay and oxygen quenching in aqueous systems. Compared to these reported studies, the phosphorescence lifetimes of our **DCorBF₂-BSA** conjugates at frozen state are among the longest in the reported studies of aqueous afterglow systems. The photophysical data for the other **BF₂bdk-BSA** conjugates are detailed in Figs. S40–S48.

Furthermore, by varying the types of proteins and using **2COOHNAPBF₂** as the luminophore, we successfully labeled streptavidin (SA), γ-globulin, silk protein, concanavalin A, and insulin, resulting in five conjugates that emitted afterglow visible by the human eye

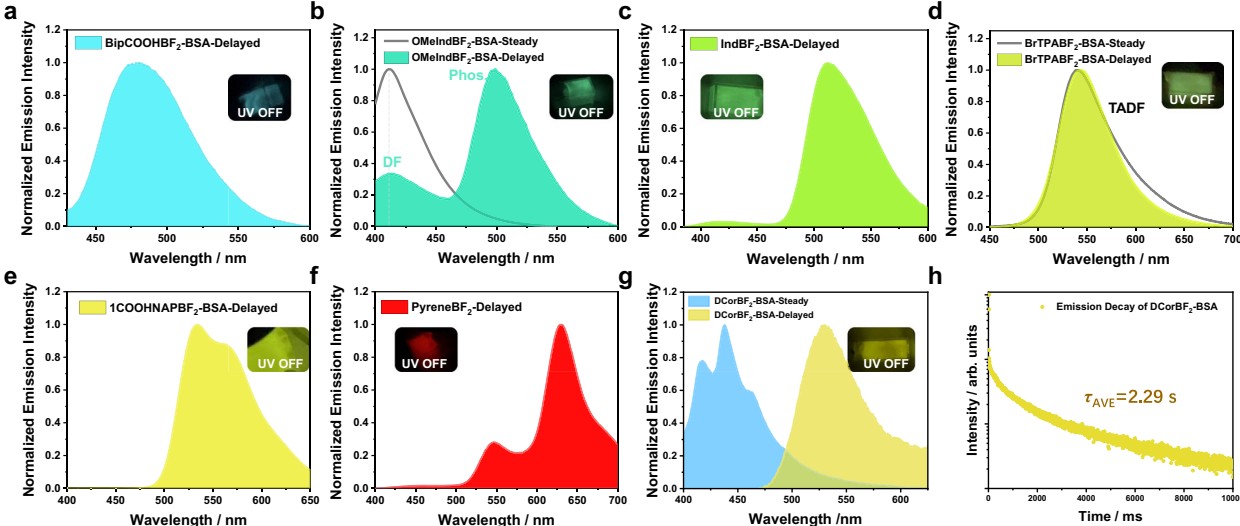

**Fig. 6 | Photophysical property of BF$_2$bdk-BSA conjugates.** Emission spectra of the afterglow ice of **BipCOOHBF$_2$-BSA** (**a**), **OMeIndBF$_2$-BSA** (**b**), **IndBF$_2$-BSA** (**c**), **BrTPABF$_2$-BSA** (**d**), **1COOHNAPBF$_2$-BSA** (**e**), **PyreneBF$_2$-BSA** (**f**) conjugates upon excitation at 365 nm. **g** Emission spectrum of the afterglow ice **DCorBF$_2$-BSA** conjugates upon excitation at 365 nm and (**h**) the corresponding emission decay curve monitored at 530 nm. Source data are provided as a Source Data file.

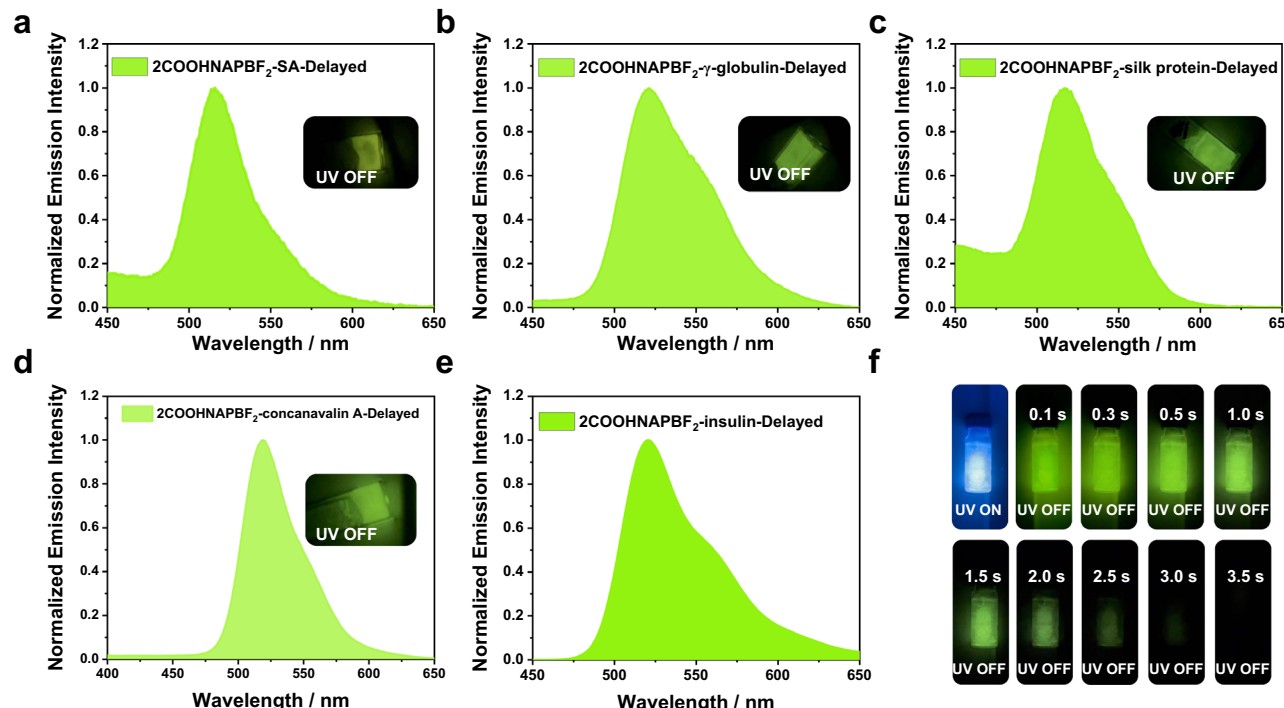

**Fig. 7 | Photophysical property of 2COOHNAPBF$_2$-protein conjugates.** Delayed emission spectra of the afterglow ice of **2COOHNAPBF$_2$-SA** (**a**), **2COOHNAPBF$_2$-γ-globulin** (**b**), **2COOHNAPBF$_2$-silk protein** (**c**), **2COOHNAPBF$_2$-concanavalin A** (**d**), **2COOHNAPBF$_2$-insulin** (**e**) conjugates upon 365 nm excitation. **f** Photographs of the afterglow ice of **2COOHNAPBF$_2$-insulin** conjugates under UV and after switching off UV excitation. Source data are provided as a Source Data file.

at freezing temperatures (Fig. 7). The emission decay curve corresponding to the five **2COOHNAPBF$_2$**-protein conjugates are presented in Supplementary Fig. 49. Among these conjugates, the **2COOHNAPBF$_2$-insulin** conjugates exhibited the brightest yellow-green phosphorescent afterglow (Fig. 7f). These results further support the versatility of β-diketone-based method for protein labeling and its ability to trigger phosphorescent emission in the resulting conjugates.

For the β-diketone-based method, the luminophors (either the β-diketone or the **BF$_2$bdk**) are relatively stable and allow long-term storage under ambient conditions. Besides, our research group builds a molecular library that now contains over 500 **BF$_2$bdk** compounds, from which we can choose luminophors with diverse phosphorescence colors and other desired properties. Moreover, most of the luminophores have intramolecular charge transfer property, which can be readily excited by UVA and visible lights, show strong

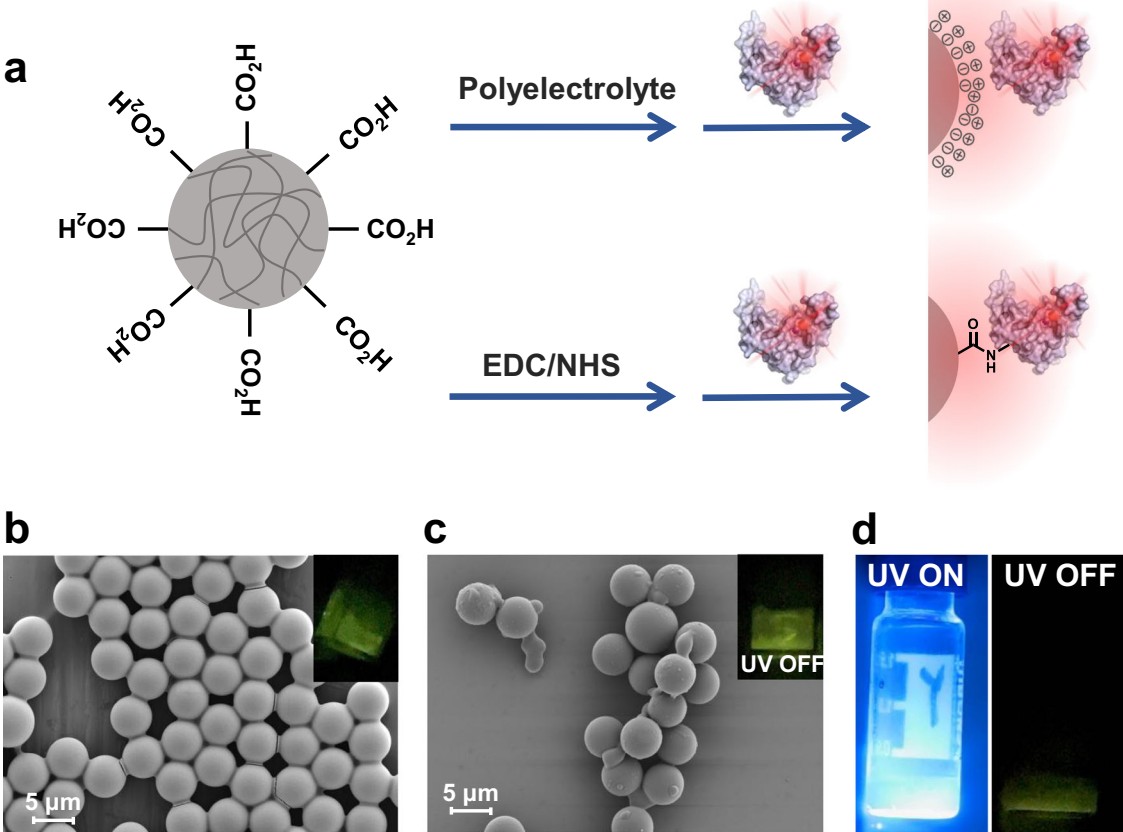

**Fig. 8 | Organic afterglow protein microspheres. a** Schematic illustration of the preparation of organic afterglow protein microspheres. **b** SEM images of organic afterglow **2COOHNAPBF₂-BSA** microspheres prepared using polyethyleneimine. **c** SEM images of organic afterglow **2COOHNAPBF₂-BSA** microspheres prepared using poly-L-arginine. For (**b**, **c**) two times each SEM experiment was repeated independently with similar results. In each time, four SEM images were taken and the images shown are representative of these repeats. **d** Photographs (under UV and upon ceasing UV excitation) of the frozen organic afterglow microspheres modified by **2COOHNAPBF₂-insulin** conjugates via NHS ester chemistry.

intersystem crossing efficiency and modest phosphorescence decay rate. Consequently, the resultant luminophore-protein conjugates would exhibit significant phosphorescence property at freezing temperature.

## Organic afterglow protein microspheres

Given the promising applications of luminescent microspheres in the analysis and detection of biomacromolecules[73–76], we conducted a study on the modification of polymer microspheres with luminophore-protein conjugates (Fig. 8a). Specifically, we selected monodisperse carboxylated polystyrene microspheres with a size of 5 micrometers. Initially, the microspheres were coated with positively charged polyethyleneimine to impart a positive surface charge. Subsequently, negatively charged **2COOHNAPBF₂-BSA** conjugates were electrostatically attracted to and anchored onto the surface of the microspheres. Under 365 nm UV lamp, the decorated microspheres emitted bright afterglow visible by the human eye, as collaborated by delayed emission spectra (Supplementary Fig. 50a). Scanning electron microscopy (SEM) analysis revealed that the decorated microspheres possessed a regular morphology and uniform size distribution (Fig. 8b). By replacing polyethyleneimine by a more biocompatible polymer, poly-L-arginine, we also successfully prepared **2COOHNAPBF₂-BSA**-decorated microspheres, which maintained excellent afterglow properties and consistent microsphere morphology (Fig. 8c), as collaborated by delayed emission spectra (Supplementary Fig. 50b). In control experiments, pristine BSA was used in the place of **2COOHNAPBF₂-**

**BSA**, the resultant microspheres retained good morphology but exhibited no afterglow (Supplementary Fig. 51). These experiments demonstrated that the observed phosphorescence in the afterglow protein microspheres originated from the luminophore-BSA conjugates. In addition to the polyelectrolyte coating method, we activated the carboxylated polystyrene microspheres with NHS ester and covalently linked the bright phosphorescent **2COOHNAPBF₂-insulin** conjugates to the microsphere surface (Fig. 8a). The resulting afterglow protein microspheres emitted bright yellow-green phosphorescence when excited with a handheld 365 nm UV lamp (Fig. 8d). These microspheres inherit the properties of organic luminophore and proteins, suggesting their potential for analyzing and detecting biomacromolecules.

## Afterglow luminophore-Protein Complex via Specific Recognition

We further investigated the properties of luminophore-protein complex systems. We selected the classic streptavidin-biotin (**SA-Biotin**) pair, renowned for its binding constant on the order of $10^{15}$ M$^{-1}$[77,78], to provide specific recognition interaction in the present system. For the molecular design, **NAPpBP** was chosen as the phosphorescent moiety because of its strong ISC and potential of long-lifetime phosphorescence. We first synthesized **NAPpBP-NH₂** and then connected it with biotin to obtain **NAPpBP-Biotin** (Fig. 9a). The corresponding phosphorescent molecule **NAPpBP-NHCOCH₃** has also been synthesized, which exhibited a bright yellow afterglow when doped in suitable

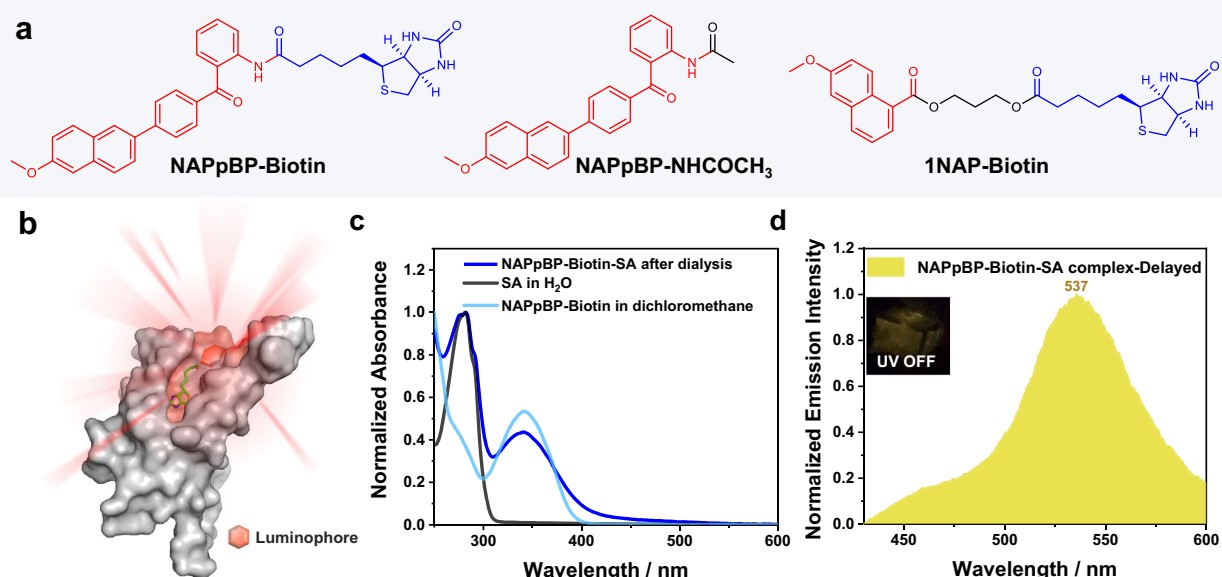

**Fig. 9 | Afterglow luminophore-protein complex via specific recognition.**
**a** Chemical structures of **NAPpBP-Biotin**, **NAPpBP-NHCOCH₃** and **1NAP-Biotin**.
**b** Schematic illustration of the afterglow luminophore-protein complex via biotin-SA specific recognition. **c** UV-vis absorption spectra of the **NAPpBP-Biotin-SA** complex after dialysis, SA aqueous solution and **NAPpBP-Biotin** in dichloromethane solution. **d** Delayed emission spectrum of the frozen **NAPpBP-Biotin-SA** complex upon excitation at 365 nm. Source data are provided as a Source Data file.

small-molecule matrix and excited at 365 nm (Supplementary Fig. 52). According to the reported studies[79,80], chemical linkage of functional moiety to biotin's carboxylic acid group would not affect the specific binding property between biotin group and SA.

For the formation of luminophore-protein complex (Fig. 9b), **NAPpBP-Biotin** was added into SA aqueous solution and the obtained mixture was incubated at room temperature for 2 h. After dialysis against water, the resultant solution has been found to show UV-vis absorption band in the region of 300–400 nm, which can be exclusively assigned to **NAPpBP** group (Fig. 9c). At freezing point, the solution has been found to emit a yellow afterglow upon 365 nm excitation. Its delayed emission spectrum revealed a phosphorescence band at 537 nm (Fig. 9d), similar to that of **NAPpBP-NHCOCH₃** sample in doped state (Supplementary Fig. 52). In control experiments, when **NAPpBP-Biotin** was not added, the frozen solution of pure SA didn't show afterglow (Supplementary Fig. 53). The frozen solution of **Biotin-SA** also didn't show afterglow (Supplementary Fig. 54). **NAPpBP** was also modified with PEG and the aqueous solution of **NAPpBP-PEG** at 10 mg/mL (**NAPpBP** group, about 1.49 mg/mL) at frozen state showed no afterglow (Supplementary Fig. 55). In contrast, a clear afterglow was observed for the **NAPpBP-Biotin-SA** complex at 1 mg/mL (**NAPpBP** group, about 0.088 mg/mL), confirming the significance of the protein's hydrophobic microregion for afterglow emergence. Furthermore, when we replaced **NAPpBP-Biotin** with **NAPpBP-NHCOCH₃**, and subjected it to the same experimental conditions with SA, no afterglow was produced (Supplementary Fig. 56). Similarly, treatment of **NAPpBP-Biotin** with BSA under identical conditions resulted in only a very weak afterglow (Supplementary Fig. 57). These experiments demonstrate that **NAPpBP-Biotin** can specifically bind to SA, forming the **NAPpBP-Biotin-SA** complex. Within this complex, the T₁ state of **NAPpBP** moiety upon UV excitation can be effectively protected by the hydrophobic microregion of SA, allowing for the bright afterglow emission. The **NAPpBP-Biotin-SA** complex (0.5 mg/mL to 2.0 mg/mL) has negligible cytotoxicity to NIH3T3 cells (Supplementary Fig. 58). Moreover, we also synthesized **1NAP-Biotin** to complex with SA under the same experimental conditions; the **1NAP** group has weak phosphorescent property compared to **NAPpBP**. The resultant **1NAP-Biotin-SA** complex in the frozen state yielded no afterglow

(Supplementary Fig. 59), highlighting the importance of strong phosphorescent units for the formation of afterglow luminophore-protein complex. Since free **NAPpBP** in aqueous system didn't show afterglow at freezing temperature (as can be seen in the case of **NAPpBP-PEG**, Supplementary Fig. 55), we can simplify the experimental procedure by directly adding powders of **NAPpBP-Biotin** into aqueous solution of streptavidin. After stirring the mixture for 8 h, the **NAPpBP-Biotin** powders disappear, which suggest the **NAPpBP-Biotin** molecules gradually dissolved in the protein solution driven by the strong **SA-Biotin** binding. After being stored at −20 °C, the frozen **NAPpBP-Biotin-SA** sample also exhibit significant afterglow (Supplementary Fig. 60). This protocol avoids the use of organic solvent and eliminate separation procedures. Based on these findings, we propose that phosphorescent molecules modified with specific moieties possess significant potential for the specific recognition of proteins. To the best of our knowledge on the function and application of organic afterglow materials, our study represents an interesting example of specific recognition and turn-on sensing of biomacromolecules by organic afterglow emitters. Despite the situation that the technology of such specific recognition and sensing is not mature, it still represents the initial step of organic afterglow materials towards corresponding biological application fields.

During revising this manuscript, we discuss with our corporate partners and perform more studies regarding the function or application of the current finding. One of our corporate partners, Shandong Longchang Animal Health Product Co., Ltd., which is a global leader in the bile acids industry, is expanding the application scenarios of its bile acid products in the biomedical field. A critical task is to identify target proteins whose activity can be controlled by bile acids. Here we build a model compound named **NAPBP-CDCA** by covalently linking chenodeoxycholic acid (CDCA) to **NABP** phosphorescent moiety (Fig. 10a); the synthetic details and structural characterization results of the **NAPBP-CDCA** can be found in Supplementary Information. After incubation of **NAPBP-CDCA** and lipase (porcine pancreas) in aqueous solution at room temperature for 8 h, the resultant mixture at freezing temperature has been found to exhibit significant afterglow that originates from the long-lived phosphorescence of **NAPBP** moiety (Fig. 10b–d). In control experiment, neither the aqueous suspension of

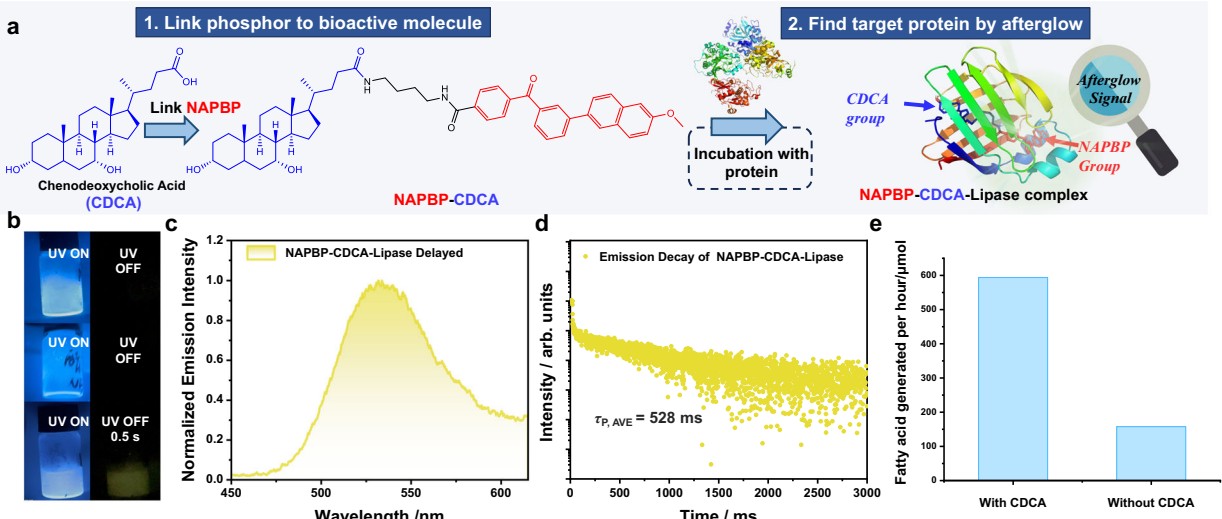

**Fig. 10 | Application of afterglow ice to identify target proteins. a** Schematic diagram of the application of afterglow ice to identify target proteins whose activity can be controlled by bile acids. **b** Photographs of the frozen aqueous suspension of **NAPBP** compounds (top), the frozen mixture of **NAPBP** and lipase in aqueous medium (middle), and the frozen solution of **NAPBP-CDCA-Lipase** complex (bottom) under UV and after switching off UV excitation. **c** Delayed emission spectra (1 ms delay) of the frozen solution of **NAPBP-CA-Lipase** complex and **d** the corresponding emission decay curve under 365 nm excitation. Source data are provided as a Source Data file. **e** Lipase activity in the presence of CDCA and in the absence of CDCA.

**NAPBP** compounds nor the mixture of **NAPBP** and lipase in aqueous medium show afterglow at freezing temperature (Fig. 10b). With reference to the results in **NAPBP-Biotin-SA** system, such observation in **NAPBP-CDCA-lipase** system suggest some non-covalent association, binding or recognition between CDCA and lipase; in this way, **NAPBP** moiety would deposit in some hydrophobic domain of lipase and its $T_1$ state can be well protected by lipase at freezing temperature, which can explain the emergence of significant afterglow in this **NAPBP-CDCA-lipase** system. With this information, we test lipase activity in the presence of CDCA. It has been found that 594 μmol of fatty acid can be generated from olive oil per hour at 37 °C and at pH 8.3 (5% NaHCO₃) by using 1 mg/mL lipase in the presence of 10 mg/mL CDCA (Fig. 10e). Such amount of fatty acid generated is much larger than that in the control experiment without the addition of CDCA (158 μmol). These results not only demonstrate that CDCA can significantly enhance lipase activity (which is very important for animal health and growth) but also indicate that our technology based on afterglow ice provides a visual pathway to identify target proteins for specific biologically active compounds. Further studies are still in progress, which would enlarge the application scope of bile acids in biomedical areas in the near future.

## Discussion

This study presents the intriguing afterglow ice formed by the aqueous solution of luminophore-protein conjugates and complexes at freezing temperature. Hydrophobic interaction and covalent linkage/strong recognition interaction between luminophore and protein have been found to protect luminophore's triplet excited states, leading to the emergence of long-lived phosphorescence in the afterglow ice system. In control experiment, free luminophore shows no afterglow at freezing temperature. Based on these findings, we can envisage the emergence of interesting techniques such as no-wash afterglow labeling of protein, as well as turn-on afterglow probe for specific recognition of protein. All these techniques can avoid background fluorescence interference, would exhibit high signal-to-noise ratio, and would become important methods for bio-labeling and bioassays. Further studies would be placed on these directions.

This study reports the interesting example of organic afterglow protein microspheres and the instance of targeted recognition and sensing of biomacromolecules utilizing organic afterglow emitters.

Although it is not yet a mature technology for biomedical applications, this study on organic afterglow materials represents the initial step towards bio-labeling and bioassay fields, as well as expanding the application scenarios of bioactive products.

## Methods

### Preparation of organic RTP materials by doping with BA matrix

For the preparation of phosphor-BA-0.1% materials, 100 μL phosphorescent molecules in acetone (1 mg/mL) and 100 mg benzoic acid (BA) were first added to an agate mortar at a weight ratio of 1:1000 and were mixed by grinding with the aid of a mixed solvent of acetone-ethanol. The mixed solvent of acetone-ethanol, which can dissolve both phosphorescent molecules and BA, assists the mixing of phosphorescent molecules and BA. After grinding and solvent evaporation, the organic two-component afterglow materials was obtained.

### Preparation of luminophore-protein conjugates

In an 8 mM sodium bicarbonate solution, the phosphorescent molecule was mixed with the protein. The resulting mixture was stirred at 25 °C for 2 h to ensure adequate interaction between the phosphorescent molecule and the protein. Following the incubation period, the mixture was subjected to dialysis to remove any unbound phosphorescent molecules and other low molecular weight impurities. The dialysis was conducted over a period of 48 h, during which the solution was dialyzed four times, each time against fresh deionized water. This process allowed for the gradual purification of the luminophore-protein conjugates, yielding a highly purified dialysate containing the desired complex. The resulting luminophore-protein conjugates was then collected.

### Preparation of organic afterglow protein microspheres assisted by polyelectrolyte binder

The polyelectrolyte (either polyethyleneimine or poly-L-arginine) and carboxylated PS microspheres were stirred at 25 °C for 30 min to facilitate the modification of the microsphere surface with the polyelectrolyte. Afterward, the reaction mixture was subjected to centrifugation to remove the supernatant, which contained any unbound polyelectrolytes. The polyelectrolyte-modified microspheres were then immediately resuspended in a solution containing

**2COOHNAPBF$_2$-BSA** conjugates. The suspension was allowed to stand at room temperature for an additional 30 min, facilitating the electrostatic attachment of the **2COOHNAPBF$_2$-BSA** conjugate to the microsphere surface. To ensure the removal of any unreacted conjugates, the reaction mixture was subjected to four to five rounds of centrifugation. The obtained organic afterglow protein microspheres were then dispersed in water by using Pluronic F127, a triblock copolymer, to achieve a stable colloidal suspension.

## Preparation of organic afterglow protein microspheres via NHS ester chemistry

*N*-Hydroxysuccinimide (NHS), 3-(3-dimethylaminopropyl)-1-ethylcarbodiimide hydrochloride (EDC) and carboxylated microspheres were stirred at 25 °C for 30 min to facilitate the modification of the microsphere surface with NHS ester. After the activation step, the reaction mixture was subjected to centrifugation to remove the supernatant, which contained any unreacted reagents. The activated microspheres were then immediately resuspended in a solution containing **2COOHNAPBF$_2$-insulin** conjugates. The suspension was allowed to react at room temperature for an additional 30 min, facilitating the covalent attachment of the **2COOHNAPBF$_2$-insulin** conjugate to the microsphere surface. To ensure the removal of any unreacted conjugates, the reaction mixture was subjected to four to five rounds of centrifugation. The obtained organic afterglow protein microspheres were then dispersed in water by using Pluronic F127, a triblock copolymer, to achieve a stable colloidal suspension.

## Preparation of Luminophore-Protein Complexes

In an 8 mM sodium bicarbonate solution, the phosphorescent molecule was mixed with the protein. The resulting mixture was stirred at 25 °C for 2 h to ensure adequate interaction between the phosphorescent molecule and the protein. Following the incubation period, the mixture was subjected to dialysis to remove any unbound phosphorescent molecules and other low molecular weight impurities. The dialysis was conducted over a period of 13 h, during which the solution was dialyzed four times, each time against fresh deionized water. This process allowed for the gradual purification of the luminophore-protein complex, yielding a highly purified dialysate containing the desired complex. The resulting luminophore-protein complex was then collected.

## Reporting summary

Further information on research design is available in the Nature Portfolio Reporting Summary linked to this article.

## Data availability

The data that support the findings of this study are available in the article, supplementary information file, source data file, and from the corresponding author on request. The X-ray crystallographic coordinates for structures reported in this study have been deposited at the Cambridge Crystallographic Data Centre (CCDC), under deposition number 2505526. These data can be obtained free of charge from The Cambridge Crystallographic Data Centre via www.ccdc.cam.ac.uk/data_request/cif. The atomic coordinates of the optimized ground states of **NAPBP-COOH** are provided in Supplementary Data 1. Source data are provided with this paper.

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

## Acknowledgements

We thank the financial supports from National Natural Science Foundation of China (22475228, 22175194, K.Z.), the Strategic Priority Research Program of the Chinese Academy of Sciences (XDB0610000, K.Z.), Hundred Talents Program from Shanghai Institute of Organic Chemistry (Y121078, K.Z.), Pioneer Hundred Talents Program of Chinese Academy of Sciences (E320021, K.Z.), and Ningbo Natural Science Foundation (2023J243, K.Z.). We acknowledge the assistance of Mr. Hongxin Gao, Mr. Biao Xu, Mr. Zi Ye and Miss Yan Jiang in the experiments.

## Author contributions

X.L. and K.Z. conceived the idea and designed the experiments. X.L., J.L., G.W., Y.S. and M.W. performed the experimental studies and carried out the analysis. X.L. and K.Z. co-wrote the manuscript. K.Z. directed the project.

## Competing interests

The authors declare no competing interests.
