## [Transparent Peer Review file · Nature Communications]

Afterglow ice formed by phosphorescent luminophore-protein conjugates and complexes in aqueous solution at freezing temperature

Corresponding Author: Professor Kaka Zhang

Version 0:

Reviewer comments:

Reviewer #1

(Remarks to the Author)

This manuscript reports an interesting study on the phenomenon termed afterglow ice, representing organic phosphorescence from aqueous solutions of luminophore-protein conjugates and complexes at freezing temperatures. The authors utilized established labeling methods, including NHS ester chemistry and innovative β -diketone-protein conjugation, to demonstrate the substantial afterglow emission. This work broadens the functional applicability of organic phosphorescence to bioanalytical contexts, introducing specific recognition and sensing of biomacromolecules by organic afterglow emitters. Although preliminary in practical application, the manuscript provides rigorous experiments, detailed controls, and insightful mechanistic explorations. I thus suggest it for publication in Nature Communications following some revisions. My detailed comments and suggestions are listed below:

- (1) Benzophenone is a well-established ISC-promoting moiety, and its incorporation into a molecular system typically enhances the intersystem crossing quantum yield while reducing the phosphorescence lifetime. However, in this study, the introduction of benzophenone into naphthoic acid results in a prolonged phosphorescence lifetime. The authors are encouraged to provide a more comprehensive explanation for this phenomenon.
- (2) The authors should provide an assessment of the biotoxicity of the luminophore-protein conjugates or complexes in this study.
- (3) The authors have developed a protein labeling method based on β -diketone molecules. I would appreciate a more detailed explanation of the advantages of this approach relative to conventional NHS ester labeling methods. A comparative analysis supported by experimental data or literature references would significantly strengthen the manuscript by highlighting the unique contributions and potential benefits of this new labeling strategy.
- (4) While the authors acknowledge that this technology is still immature for immediate biomedical applications, a more explicit discussion outlining the steps necessary to bridge current findings to practical bioanalytical methods would enrich the manuscript.

Reviewer #2

(Remarks to the Author)

This manuscript presents a pioneering concept—to the best of my knowledge, the first demonstration of organic phosphorescence (“afterglow”) from aqueous solutions of luminophore-protein conjugates and complexes at sub-zero temperatures. The authors aim to explore organic afterglow emissions in biomacromolecular systems and introduce a novel β -diketone-based chemical labeling approach. The combination of hydrophobic interactions, covalent bonding, and low-temperature stabilization is proposed to explain the observed afterglow phenomena. The core of the work focuses on synthesizing new luminophores that can be covalently attached to proteins. The authors then carry out a well-designed set of control experiments to demonstrate that the luminophore tags are indeed covalently

bound. The successful observation of long-lived phosphorescence is impressive and holds clear potential for future applications—congratulations!

However, at this stage, the findings remain somewhat heuristic. While the manuscript presents compelling visual and spectroscopic evidence (e.g., UV-Vis, phosphorescence spectra, and control experiments) supporting the formation of luminophore–protein conjugates, it lacks direct, quantitative, or structural evidence for covalent or specific binding between the luminophores and the proteins. Moreover, the role of temperature and the physical state (i.e., the frozen aqueous environment) in enabling or enhancing phosphorescence is not sufficiently explored. Temperature-dependent measurements of afterglow lifetimes would strengthen the argument that the matrix (ice) plays an essential role. Apparently, the phosphorescence is related to actual conjugate formation but what are the other necessary conditions induced by low temperature? Could it be precipitation or phase separation?

Lastly, the term “afterglow ice” may be misleading, as it implies that the ice itself is the source of luminescence. A more precise term could better reflect the actual origin of the observed phenomenon.

Reviewer #3

(Remarks to the Author)

In this paper, Li et al. report an organic phosphorescence phenomenon from a series of luminophore-protein conjugates which show afterglow emission at frozen temperature and they termed afterglow ice for abbreviation. In addition, they also showed that β -diketone moiety can be used to label protein via the formation of enamine chemistry. While the authors have synthesized a series of fluorescent dye to showcase the phenomenon, I do not believe that the work is suitable for Nature Communications. The reasons are following;

1. The potential applications of the afterglow ice phenomenon seem very limited, either in chemical sensing, therapeutics or others. As most of the biochemistry and chemical sensing are conducted at 37 C or room temperature. The authors should showcase one example that can explore the afterglow ice observation.
2. The authors should compare the delayed emission spectra of the luminophore-protein conjugates at room temperature and at the frozen temperature.
3. The authors should perform a temperature titration experiment to understand the temperature effect and describe in detail the mechanism. For example, what is the cause of this afterglow effect, restricted bond rotation in macromolecules, hydrophobic effect or temperature.
4. The quantitative analysis of the afterglow ice luminophore-protein conjugates should be compared with other afterglow materials, not just only the time-resolved analysis.
5. Protein labeling using β -diketone moiety should be compared with the NHS ester group to compare the labeling efficiency. Furthermore, the SDS-PAGE gel shown in Figure 3d is not very convincing due to the strong fluorescence background.
6. The interaction of NAPpBP-Biotin with streptavidin to generate afterglow signals have to undergo separation and performed at the frozen temperature. While I understand that this is just a proof-of-principle demonstration, the protocol has limited the potential application of the molecular interaction.

Reviewer #4

(Remarks to the Author)

Version 1:

Reviewer comments:

Reviewer #1

(Remarks to the Author)

The authors have carefully addressed all the concerns of the reviewers. This version is acceptable for publication.

Reviewer #3

(Remarks to the Author)

The authors have adequately addressed the reviewers' concerns. I recommend that the article be accepted for publication.

Reviewer #4

(Remarks to the Author)

[Editorial Note: Please see end of file for full review]

I co-reviewed this manuscript with one of the reviewers who provided the listed reports. This is part of the Nature

Communications initiative to facilitate training in peer review and to provide appropriate recognition for Early Career Researchers who co-review manuscripts.

Version 2:

Reviewer comments:

Reviewer #4

(Remarks to the Author)

[Editorial Note: Please see end of file for full review]

REVIEWER COMMENTS

Reviewer #1 (Remarks to the Author):

This manuscript reports an interesting study on the phenomenon termed afterglow ice, representing organic phosphorescence from aqueous solutions of luminophore-protein conjugates and complexes at freezing temperatures. The authors utilized established labeling methods, including NHS ester chemistry and innovative β -diketone-protein conjugation, to demonstrate the substantial afterglow emission. This work broadens the functional applicability of organic phosphorescence to bioanalytical contexts, introducing specific recognition and sensing of biomacromolecules by organic afterglow emitters. Although preliminary in practical application, the manuscript provides rigorous experiments, detailed controls, and insightful mechanistic explorations. I thus suggest it for publication in Nature Communications following some revisions. My detailed comments and suggestions are listed below:

Response: We thank the reviewer for the very positive comments on our manuscript. Since the detailed suggestions are listed below, we make corresponding response in the following.

(1) Benzophenone is a well-established ISC-promoting moiety, and its incorporation into a molecular system typically enhances the intersystem crossing quantum yield while reducing the phosphorescence lifetime. However, in this study, the introduction of benzophenone into naphthoic acid results in a prolonged phosphorescence lifetime. The authors are encouraged to provide a more comprehensive explanation for this phenomenon.

Response 1: We thank the reviewer for the valuable comments and suggestion. Benzophenone (BP) is recognized as a significant phosphorescent emitter, characterized by an intersystem crossing efficiency that approaches unity. However, it exhibits a notably short phosphorescence lifetime, typically ranging from 0.1 to 1 ms, primarily due to the pronounced $n-\pi^*$ transition inherent to its triplet state. Recent research indicates that advanced charge transfer technology can maintain strong intersystem crossing in benzophenone-derived organic systems and simultaneously reduce phosphorescence rate constant (k_p) to small values^[23,70]. When nonradiative decay pathways are restricted using a rigid environment, it becomes possible to achieve highly efficient and persistent

room-temperature phosphorescence. The benzophenone moiety is electron-deficient. By connecting it with electron-donating aromatic groups, an intramolecular charge transfer (ICT) system is established. The donor group contributes actively to the triplet excited state formation of the resulting system. If the donating group possesses a low-energy T_1 state of localized excitation (LE) character, the T_1 state of the resulting molecules would consist of remarkable portion from the donating group's T_1 state. As a result, the resulting molecules' T_1 states would inherit the donating group's LE character and thus have small k_p . Naphthoic acid features a relatively low T_1 energy level; upon conjugation with the BP group, the resultant **NAPBP-COOH** molecule capitalizes on the localized excitation characteristics of naphthoic acid's triplet state, thereby facilitating the formation of a long-lived T_1 state within **NAPBP-COOH** (Supplementary Fig. 18). This molecular design effectively preserves the robust intersystem crossing capability of the benzophenone framework while attenuating the contribution of the $n-\pi^*$ transition to the T_1 state (Supplementary Fig. 18). Consequently, this configuration endows the luminophore with both high intersystem crossing efficiency and a small phosphorescence rate constant. The above explanation on the long phosphorescence lifetime of **NAPBP-COOH** system has been added in the main text of this revised manuscript.

(2) The authors should provide an assessment of the biotoxicity of the luminophore-protein conjugates or complexes in this study.

Response 2: We thank the reviewer for the valuable suggestion. Accordingly, we perform MTT cytotoxicity assays in this revised manuscript, where the experimental details have been added in this revised supplementary information. **NAPBP-BSA** conjugates, **2COOHNAP-diketone-BSA** conjugates, **NAPBP-Biotin-SA** complex have been selected for the cytotoxicity measurement (Supplementary Fig. 21, 34 and 58). The samples at low concentrations (2.5 mg/mL to 7.5 mg/mL) have negligible or low cytotoxicity to NIH3T3 cells. The cytotoxicity increases with sample concentration but is still acceptable at higher concentrations (10 mg/mL). These results have been added in this revised manuscript.

(3) The authors have developed a protein labeling method based on β -diketone molecules. I would appreciate a more detailed explanation of the advantages of this approach relative to conventional

NHS ester labeling methods. A comparative analysis supported by experimental data or literature references would significantly strengthen the manuscript by highlighting the unique contributions and potential benefits of this new labeling strategy.

Response 3: We thank the reviewer for the valuable suggestion. It is known that conventional NHS esters are moisture-sensitive and are recommended to be stored under an inert atmosphere. For the β -diketone-based method, the luminophors (either the β -diketone or the BF₂bdk) are relatively stable and allow long-term storage under ambient conditions. Besides, our research group builds a molecular library that now contains over 500 BF₂bdk compounds, from which we can choose luminophors with diverse phosphorescence colors and other desired properties. Moreover, most of the luminophores have intramolecular charge transfer property, which can be readily excited by UVA and visible lights, show strong intersystem crossing efficiency and modest phosphorescence decay rate. Consequently, the resultant luminophore-protein conjugates would exhibit significant phosphorescence property at freezing temperature. These contents have been added in the main text of the revised manuscript.

(4) While the authors acknowledge that this technology is still immature for immediate biomedical applications, a more explicit discussion outlining the steps necessary to bridge current findings to practical bioanalytical methods would enrich the manuscript.

Response 4: We thank the reviewer for the valuable suggestion. We understand that this comment refers to the function or application of the current finding. Accordingly, we perform more studies in this revised manuscript. Our corporate partner, Shandong Longchang Animal Health Product Co., Ltd., which is a global leader in the bile acids industry, is expanding the application scenarios of its bile acid products in the biomedical field. A critical task is to identify target proteins whose activity can be controlled by bile acids. Here we build a model compound named **NAPBP-CDCA** by covalently linking chenodeoxycholic acid (CDCA) to **NAPBP** phosphorescent moiety; the synthetic details and structural characterization results of the **NAPBP-CDCA** can be found in Supplementary Information. After incubation of **NAPBP-CDCA** and lipase (porcine pancreas) in aqueous solution at room temperature for 8 h, the resultant mixture at freezing temperature has been found to exhibit significant afterglow that originates from the long-lived phosphorescence of **NAPBP** moiety (Fig. 9b, c and d). In control experiment, neither the aqueous suspension of **NAPBP** compounds nor the

mixture of **NAPBP** and lipase in aqueous medium show afterglow at freezing temperature (Fig. 9b). With reference to the results in **NAPpBP-Biotin-SA** system, such observation in **NAPBP-CDCA-lipase** system suggest some non-covalent association, binding or recognition between CDCA and lipase; in this way, **NAPBP** moiety would deposit in some hydrophobic domain of lipase and its T₁ state can be well protected by lipase at freezing temperature, which can explain the emergence of significant afterglow in this **NAPBP-CDCA-lipase** system. With this information, we test lipase activity in the presence of CDCA. It has been found that 594 μmol of fatty acid can be generated from olive oil per hour at 37 ° C and at pH 8.3 (5% NaHCO₃) by using 1 mg/mL lipase in the presence of 10 mg/mL CDCA (Fig. 9e). Such amount of fatty acid generated is much larger than that in the control experiment without the addition of CDCA (158 μmol). These results not only demonstrate that CDCA can significantly enhance lipase activity (which is very important for animal health and growth) but also indicate that our technology based on afterglow ice provides a visual pathway to identify target proteins for specific biologically active compounds. Further studies are still in progress, which would enlarge the application scope of bile acids in biomedical areas in the near future. These contents have been added in the main text of this revised manuscript.

Reviewer #2 (Remarks to the Author):

This manuscript presents a pioneering concept—to the best of my knowledge, the first demonstration of organic phosphorescence (“afterglow”) from aqueous solutions of luminophore–protein conjugates and complexes at sub-zero temperatures. The authors aim to explore organic afterglow emissions in biomacromolecular systems and introduce a novel β -diketone-based chemical labeling approach. The combination of hydrophobic interactions, covalent bonding, and low-temperature stabilization is proposed to explain the observed afterglow phenomena.

The core of the work focuses on synthesizing new luminophores that can be covalently attached to proteins. The authors then carry out a well-designed set of control experiments to demonstrate that the luminophore tags are indeed covalently bound. The successful observation of long-lived phosphorescence is impressive and holds clear potential for future applications—congratulations!

Response: We thank the reviewer for the very positive and enthusiastic comments. Since the

detailed comments and suggestion are listed below, we make corresponding response in a point-to-point manner in the following.

Comment 1: However, at this stage, the findings remain somewhat heuristic. While the manuscript presents compelling visual and spectroscopic evidence (e.g., UV-Vis, phosphorescence spectra, and control experiments) supporting the formation of luminophore–protein conjugates, it lacks direct, quantitative, or structural evidence for covalent or specific binding between the luminophores and the proteins.

Response 1: We thank the reviewer for the valuable comments and suggestion. Accordingly, we perform matrix-assisted laser desorption/ionization time-of-flight (MALDI-TOF) mass spectrometry for pristine BSA and **1NAP-BSA** conjugates. The average molecular weight of **1NAP-BSA** conjugates (66960 Da) exceeds that of BSA (66531) by 429 Da (Supplementary Fig. 1), corresponding to the addition of approximately two **1NAP** group. These results agree well with the feeding molar ratio of **1NAP-NHS/BSA** (2/1) and the estimated labeling efficiency (95.6%) (regarding the details of labeling efficiency, please find the response to Reviewer 3's Comment 5). It is known that MALDI-TOF works at relatively high energy, where weak bonds and non-covalent binding would be disrupted, so the molecular increase by 429 Da can be unambiguously attributed to the covalent linkage of **1NAP** group to BSA through NHS ester chemistry.

Regarding the covalent linkage between β -diketone and BSA, we use various small-molecule amines as models to test the chemistry. We divided **2COOHNAPBF₂** into five groups under alkaline conditions: one untreated group and the remaining groups treated with various amines, including glycine (Gly), L-lysine (Lys), L-arginine (L-Arg), and n-butylamine. High-performance liquid chromatography with mass spectrometry (HPLC-MS) was employed to monitor the formation of β -diketone and β -enaminone products. The HPLC-MS results indicated that free **2COOHNAPBF₂** generates a β -diketone with a m/z of 287 in mildly alkaline conditions (Supplementary Fig. 28). Additionally, β -diketone peaks and corresponding β -enaminone peaks were observed for all four amine-treated systems (Supplementary Fig. 29 and S30), confirming that β -diketones can react with amine groups to produce β -enaminones. Moreover, we aimed to further establish the feasibility of β -diketone-based protein labeling through SDS-PAGE electrophoresis. Following reactions of five **BF₂bdk** compounds with BSA in alkaline conditions and subsequent dialysis, we obtained five

BF₂bdk-BSA aqueous solutions exhibiting fluorescence when excited at 365 nm. These were tested alongside a control group of pristine BSA in SDS-PAGE. All six samples exhibited similar migration patterns under daylight. Under 365 nm excitation, the **BF₂bdk**-BSA solutions showed fluorescence, while the pristine BSA didn't exhibit any fluorescence (Fig. 3d). This result corroborates the ability of β -diketones to achieve covalent labeling of proteins.

These contents have been highlighted in the main text of this revised manuscript.

Comment 2: Moreover, the role of temperature and the physical state (i.e., the frozen aqueous environment) in enabling or enhancing phosphorescence is not sufficiently explored. Temperature-dependent measurements of afterglow lifetimes would strengthen the argument that the matrix (ice) plays an essential role.

Response 2: We thank the reviewer for the valuable comments and suggestion. Accordingly, we collect temperature-dependent delayed emission spectra and emission decay profiles for **NAPBP-BSA** and **2COOHNAP-diketone-BSA** conjugates (Supplementary Fig. 17 and 25). The delayed emission intensity is very weak at 278 K because the active nonradiative decay and oxygen quenching of triplet excited states in the fluidic aqueous solution. Upon freezing, the delayed emission intensity has significant increase, indicating the essential role of ice for the protection of luminophore's triplet excited states. It has also been found that, by further decreasing the temperature, the delayed emission intensity and phosphorescence lifetimes increase, which can be attributed to better protection of luminophore's triplet excited states within the ice at lower temperature. These contents have been added in this revised manuscript.

Comment 3: Apparently, the phosphorescence is related to actual conjugate formation but what are the other necessary conditions induced by low temperature? Could it be precipitation or phase separation?

Response 3: We thank the reviewer for the valuable comments. We understand that some compounds may precipitate at low temperature. In this revised manuscript, the aqueous solution of **NAPBP-BSA** conjugates are immediately frozen by liquid nitrogen and then placed in -20 °C refrigerator for one day. The sample shows homogenous afterglow upon removing excitation source (Supplementary Fig. 19). Besides, the **NAPBP-BSA** conjugates have been diluted to 1 mg/mL. The

frozen solution of this diluted sample still shows significant afterglow (Supplementary Fig. 20). Since precipitation or phase separation would unlikely occur during rapid freezing and for samples at low concentration, these results can rule out afterglow mechanism by precipitation or phase separation, and thus support our proposed mechanism of synergistic effects of conjugate formation and hydrophobic interactions at freezing temperature for the emergence of afterglow. These results and discussion have been added in this revised manuscript.

Comment 4: Lastly, the term “afterglow ice” may be misleading, as it implies that the ice itself is the source of luminescence. A more precise term could better reflect the actual origin of the observed phenomenon.

Response 4: We thank the reviewer for the valuable comments. We respectfully acknowledge the concern regarding potential ambiguity in the term “afterglow ice.” However, we would like to clarify that this term is explicitly introduced and clearly defined in the abstract of our manuscript, where we state: “the first observation of organic afterglow from aqueous solutions of luminophore-protein conjugates and complexes at freezing temperature, named as afterglow ice for abbreviation.” This phrasing makes it clear that the luminescence originates from the luminophore-protein systems embedded within the ice matrix, rather than the ice itself. Throughout the manuscript, the term “afterglow ice” is consistently used to refer to this specific phenomenon. Given that the term is clearly defined at the beginning, we believe it does not introduce any significant risk of misinterpretation. Moreover, as a novel phenomenon, “afterglow ice” serves not only as a descriptive term but also as a concise label for the breakthrough itself—facilitating clear communication of our findings within the scientific community. These contents have been added as Supporting Text S1 in Supplementary Information.

Dominik Heger and Lukáš Veselý

Reviewer #3 (Remarks to the Author):

In this paper, Li et al. report an organic phosphorescence phenomenon from a series of luminophore-protein conjugates which show afterglow emission at frozen temperature and they termed afterglow

ice for abbreviation. In addition, they also showed that β -diketone moiety can be used to label protein via the formation of enamine chemistry. While the authors have synthesized a series of fluorescent dye to showcase the phenomenon, I do not believe that the work is suitable for Nature Communications. The reasons are following;

Response: We thank the reviewer for the valuable comments. Since the detailed comments are listed below, we make corresponding response in the following.

Comment 1. The potential applications of the afterglow ice phenomenon seem very limited, either in chemical sensing, therapeutics or others. As most of the biochemistry and chemical sensing are conducted at 37 C or room temperature. The authors should showcase one example that can explore the afterglow ice observation.

Response 1: We thank the reviewer for the valuable comments and suggestion. During revising this manuscript, we discuss with our corporate partners and perform more studies regarding the function or application of the current finding of the afterglow ice phenomenon. One of our corporate partners, Shandong Longchang Animal Health Product Co., Ltd., which is a global leader in the bile acids industry, is expanding the application scenarios of its bile acid products in the biomedical field. A critical task is to identify target proteins whose activity can be controlled by bile acids. Here we build a model compound named **NAPBP-CDCA** by covalently linking chenodeoxycholic acid (CDCA) to **NAPBP** phosphorescent moiety (Fig. 9a); the synthetic details and structural characterization results of the **NAPBP-CDCA** can be found in Supplementary Information. After incubation of **NAPBP-CDCA** and lipase (porcine pancreas) in aqueous solution at room temperature for 8 h, the resultant mixture at freezing temperature has been found to exhibit significant afterglow that originates from the long-lived phosphorescence of **NAPBP** moiety (Fig. 9b, c and d). In control experiment, neither the aqueous suspension of **NAPBP** compounds nor the mixture of **NAPBP** and lipase in aqueous medium show afterglow at freezing temperature (Fig. 9b). With reference to the results in **NAPBP-Biotin-SA** system, such observation in **NAPBP-CDCA-lipase** system suggest some non-covalent association, binding or recognition between CDCA and lipase; in this way, **NAPBP** moiety would deposit in some hydrophobic domain of lipase and its T_1 state can be well protected by lipase at freezing temperature, which can explain the emergence of significant afterglow in this **NAPBP-CDCA-lipase** system. With this information, we test lipase

activity in the presence of CDCA. It has been found that 594 μmol of fatty acid can be generated from olive oil per hour at 37 ° C and at pH 8.3 (5% NaHCO_3) by using 1 mg/mL lipase in the presence of 10 mg/mL CDCA (Fig. 9e). Such amount of fatty acid generated is much larger than that in the control experiment without the addition of CDCA (158 μmol). These results not only demonstrate that CDCA can significantly enhance lipase activity (which is very important for animal health and growth) but also indicate that our technology based on afterglow ice provides a visual pathway to identify target proteins for specific biologically active compounds. Further studies are still in progress, which would enlarge the application scope of bile acids in biomedical areas in the near future. These contents have been added in the main text of this revised manuscript.

Comment 2. The authors should compare the delayed emission spectra of the luminophore-protein conjugates at room temperature and at the frozen temperature.

Response 2: We thank the reviewer for the valuable comments and suggestion. Accordingly, we collect the delayed emission spectra (1 ms delay) of **NAPBP-BSA** and **2COOHNAP-diketone-BSA** conjugates at room temperature and frozen temperature (Supplementary Fig. 16 and 26). The aqueous solution of the conjugates at room temperature shows negligible delayed emission signals, which can be explained by the active nonradiative decay and oxygen quenching of luminophore's triplet excited states at fluidic aqueous solution. At the frozen temperature, the nonradiative decay of luminophore's triplet excited states can be suppressed within ice matrix and the oxygen diffusion can be inhibited by the ice matrix, leading to the significant phosphorescence (Supplementary Fig. 16 and 26).

Comment 3. The authors should perform a temperature titration experiment to understand the temperature effect and describe in detail the mechanism. For example, what is the cause of this afterglow effect, restricted bond rotation in macromolecules, hydrophobic effect or temperature.

Response 3: We thank the reviewer for the valuable comments and suggestion. Accordingly, we collect temperature-dependent delayed emission spectra and emission decay profiles for **NAPBP-BSA** and **2COOHNAP-diketone-BSA** conjugates (Supplementary Fig. 17 and 25). The delayed emission intensity is very weak at 278 K because the active nonradiative decay and oxygen quenching of triplet excited states in the fluidic aqueous solution. Upon freezing, the delayed

emission intensity has significant increase, indicating the essential role of ice for the protection of luminophore's triplet excited states. It has also been found that, by further decreasing the temperature, the delayed emission intensity and phosphorescence lifetimes increase, which can be attributed to better protection of luminophore's triplet excited states within the ice at lower temperature.

Besides temperature, we also investigate the contribution of hydrophobic effect on the afterglow property. Sodium dodecyl sulfonate (SDS) is known to disrupt hydrophobic interactions in protein systems. In this revised manuscript, we add SDS into the aqueous solution of **2COOHNAP-diketone-BSA** conjugates. After being stored at -20 °C, the frozen sample with the presence of SDS shows much weaker phosphorescence than the frozen sample without SDS (Supplementary Fig. 27). This observation supports our proposed mechanism that hydrophobic interaction and covalent linkage between luminophore and protein can protect organic triplet excited states from quenching.

In addition, NMR technique is used to study the mobility of luminophore in different microenvironments. When **1NAP** groups are covalently linked to PEG or ϵ -polylysine (EPL) chains, the resultant **1NAP-PEG** and **1NAP-EPL** samples in D₂O show sharp and well-resolved ¹H NMR signals in the aromatic region (Supplementary Fig. 8 and 9). The **1NAP** groups have free motion in D₂O to give sharp peaks, because the hydrophilic polymers (PEG and EPL) endow **1NAP** groups with excellent solubility in D₂O. In contrast, ¹H NMR spectrum of **1NAP-BSA** conjugates in D₂O shows the disappearance of NMR signals of **1NAP** groups in the aromatic region (Supplementary Fig. 10), which suggest the motion of **1NAP** groups is seriously restricted within BSA (*Macromolecules* **2003**, 36, 2576-2578). The hydrophobic **1NAP** groups within some hydrophobic microregion of BSA have largely restricted motion, which can significant reduce nonradiative decay of **1NAP**'s triplet excited states, leading to the emergence of afterglow at freezing temperature.

The above results and discussion that further support our proposed afterglow mechanism have been added in this revised manuscript.

Comment 4. The quantitative analysis of the afterglow ice luminophore-protein conjugates should be compared with other afterglow materials, not just only the time-resolved analysis.

Response 4: We thank the reviewer for the valuable comments and suggestion. We understand that

this reviewer encourages us to enhance the performance of afterglow ice and make a comparison with other related afterglow materials. In this revised manuscript, we apply deuteration technique to increase phosphorescence lifetimes of luminophore-protein conjugates. C-D stretching vibration is much weaker than C-H stretching vibration. Deuteration of luminophores can reduce the energy of effective nonclassical vibrational mode involved in the nonradiative transition, decrease Franck-Condon factor, and consequently reduce nonradiative decay, leading to the elongation of phosphorescence lifetime (*Adv. Funct. Mater.* **2013**, 23, 3386–3397). Deuterated CorBF₂ (**DCorBF₂**) has been prepared and used to label BSA; the synthetic details can be found in Supplementary Information. The resultant aqueous solution of **DCorBF₂-BSA** conjugates at frozen state display phosphorescence lifetime of 2.29 s, which remains very rare in the related afterglow material systems (Fig. 5g and h). Supplementary Fig. 39 summarizes the reported afterglow materials in aqueous systems including supramolecular assemblies, suspension, emulsion and hydrogels, from which one can find that it is challenging to prepare materials with long phosphorescence lifetimes because of the active nonradiative decay and oxygen quenching in aqueous systems. Compared to these reported studies, the phosphorescence lifetimes of our **DCorBF₂-BSA** conjugates at frozen state are among the longest in the reported studies of aqueous afterglow systems. These contents have been added in this revised manuscript.

Comment 5. Protein labeling using β -diketone moiety should be compared with the NHS ester group to compare the labeling efficiency. Furthermore, the SDS-PAGE gel shown in Figure 3d is not very convincing due to the strong fluorescence background.

Response 5: We thank the reviewer for the valuable comments and suggestion. Accordingly, we estimate the labeling efficiency by UV-vis technique. For the NHS ester chemistry, the labeling efficiency is higher than 95% (Supplementary Fig. 2). In the case of β -diketone-based system, the labeling efficiency is 23.1% to 89.5% (Supplementary Fig. 35). It is known that conventional NHS esters are moisture-sensitive and are recommended to be stored under an inert atmosphere. For the β -diketone-based method, the luminophores (either the β -diketone or the BF₂bdk) are relatively stable and allow long-term storage under ambient conditions. Besides, our research group builds a molecular library that now contains over 500 BF₂bdk compounds, from which we can choose luminophores with diverse phosphorescence colors and other desired properties. Moreover, most of

the luminophores have intramolecular charge transfer property, which can be readily excited by UVA and visible lights, show strong intersystem crossing efficiency and modest phosphorescence decay rate. Consequently, the resultant luminophore-protein conjugates would exhibit significant phosphorescence property at freezing temperature. These contents have been added in the main text of the revised manuscript.

Regarding the SDS-PAGE gel shown in Figure 3d, we repeat the experiments for several times during revising this manuscript. The results have been added in the revised supplementary information, from which we can see the fluorescence from **BF₂bdk**-BSA conjugates (Supplementary Fig. 31). In order to remove the background of the SDS-PAGE gel, we use low-power UV excitation source, so that the fluorescence signals of **BF₂bdk**-BSA conjugates are weak in Supplementary Fig. 31. Therefore, in the main text, we still use the original Figure 3d that exhibit significant fluorescence of **BF₂bdk**-BSA conjugates. We hope the reviewer can understand that we have tried for many times to capture the photographs of SDS-PAGE gel under UV lamp. The photographs in the main text and supplementary information are the best results at the current stage.

Comment 6. The interaction of NAPpBP-Biotin with streptavidin to generate afterglow signals have to undergo separation and performed at the frozen temperature. While I understand that this is just a proof-of-principle demonstration, the protocol has limited the potential application of the molecular interaction.

Response 6: We thank the reviewer for the valuable comments and suggestion. Since free **NAPpBP** in aqueous system didn't show afterglow at freezing temperature (as can be seen in the case of **NAPpBP**-PEG, Supplementary Fig. 55), we can simplify the experimental procedure by directly adding powders of **NAPpBP-Biotin** into aqueous solution of streptavidin. After stirring the mixture for 8 h, the **NAPpBP-Biotin** powders disappear, which suggest the **NAPpBP-Biotin** molecules gradually dissolved in the protein solution driven by the strong **SA-Biotin** binding. After being stored at -20 °C, the frozen **NAPpBP-Biotin-SA** sample also exhibit significant afterglow (Supplementary Fig. 60). This protocol avoids the use of organic solvent and eliminate separation procedures. These contents have been added in this revised manuscript. Regarding the potential application of the afterglow ice, please also refer to the response to Reviewer 3's Comment 1.

Reviewer #4 (Remarks to the Author):

Response: We thank the reviewer for the valuable comments and suggestion.

REVIEWER COMMENTS

Reviewer #1 (Remarks to the Author):

The authors have carefully addressed all the concerns of the reviewers. This version is acceptable for publication.

Response: We thank the reviewer for the positive comments and recommendation on our manuscript.

Reviewer #3 (Remarks to the Author):

The authors have adequately addressed the reviewers' concerns. I recommend that the article be accepted for publication.

Response: We thank the reviewer for the positive comments and recommendation on our manuscript.

Reviewer #4 (Remarks to the Author):

[Note from the Editor: Please see attached PDF]

Overall assessment

The authors have responded carefully to the majority of our comments. They provided new data (MALDI-TOF, HPLC-MS, SDS-PAGE) that strengthen the evidence for covalent labeling and partially added temperature-dependent emission studies clarifying the role of the frozen environment.

Remaining concern – precipitation/phase separation

However, one important issue remains insufficiently addressed. It is well established that only very small solutes can be incorporated into ice crystals, while proteins and protein–dye conjugates are excluded into brine channels during freezing, which often leads to precipitation or aggregation. The rebuttal relies on rapid freezing and dilution experiments to argue against precipitation, but these do not convincingly exclude micro- or nano-scale precipitation/phase separation. Visual “homogeneity” of luminescence is not sufficient evidence. Without direct demonstration, the mechanistic conclusion that the afterglow arises solely from “hydrophobic interaction and covalent linkage protected within ice” remains incomplete.

Recommendation

The manuscript is otherwise strong and makes an important contribution. If the authors can provide more direct evidence to rule out precipitation or at least acknowledge it as a likely contributing factor, I would be fully supportive of publication.

Response: We thank the reviewer for the valuable comments and positive recommendation on our manuscript. To further study whether protein precipitation/phase separation during freezing contributes to the emergence of afterglow, we covalently link PEG chains to **NAPBP-BSA** conjugates; the steric repulsion between PEG chains would minimize the possibility of protein aggregation especially in the case when the **NAPBP-BSA** conjugates are heavily decorated with PEG chains. After being frozen, the PEG-decorated **NAPBP-BSA** conjugates have been found to show similar afterglow properties compared to **NAPBP-BSA** conjugates (control sample, without PEG) (Supplementary Fig. 20D-Q). These studies show that protein aggregation may occur during freezing in the case of luminophore-protein conjugates but has insignificant contribution to afterglow emergence in the present system. These contents, which support our proposed mechanism of synergistic effects of conjugate formation and hydrophobic interactions at freezing temperature for the emergence of afterglow, have been added in this revised manuscript.

Overall assessment

The authors have responded carefully to the majority of our comments. They provided new data (MALDI-TOF, HPLC-MS, SDS-PAGE) that strengthen the evidence for covalent labeling and partially added temperature-dependent emission studies clarifying the role of the frozen environment.

Remaining concern – precipitation/phase separation

However, one important issue remains insufficiently addressed. It is well established that only very small solutes can be incorporated into ice crystals, while proteins and protein–dye conjugates are excluded into brine channels during freezing, which often leads to precipitation or aggregation. The rebuttal relies on rapid freezing and dilution experiments to argue against precipitation, but these do not convincingly exclude micro- or nano-scale precipitation/phase separation. Visual “homogeneity” of luminescence is not sufficient evidence. Without direct demonstration, the mechanistic conclusion that the afterglow arises solely from “hydrophobic interaction and covalent linkage protected within ice” remains incomplete.

Recommendation

The manuscript is otherwise strong and makes an important contribution. If the authors can provide more direct evidence to rule out precipitation or at least acknowledge it as a likely contributing factor, I would be fully supportive of publication.

The authors have adequately addressed the reviewers' concerns. I think that the article can be accepted for publication.